# Linnemannia elongata (Mortierellaceae) stimulates Arabidopsis thaliana aerial growth and responses to auxin, ethylene, and reactive oxygen species

Natalie Vandepol[1], Julian Liber[2,3], Alan Yocca[2], Jason Matlock[4], Patrick Edger[5], Gregory Bonito[1,6]*

1 Department of Microbiology & Molecular Genetics, Michigan State University, East Lansing, Michigan, United States of America, 2 Department of Plant Biology, Michigan State University, East Lansing, Michigan, United States of America, 3 Department of Biology, Michigan Duke University, Durham, North Carolina, United States of America, 4 Department of Entomology, Michigan State University, East Lansing, Michigan, United States of America, 5 Department of Horticulture, Michigan State University, East Lansing, Michigan, United States of America, 6 Department of Plant Soil and Microbial Sciences, Michigan State University, East Lansing, Michigan, United States of America

* bonito@msu.edu

**Data Availability Statement:** Transcriptome sequence data have been accessioned in NCBI's SRA under the BioProject PRJNA704083.

## Abstract

Harnessing the plant microbiome has the potential to improve agricultural yields and protect plants against pathogens and/or abiotic stresses, while also relieving economic and environmental costs of crop production. While previous studies have gained valuable insights into the underlying genetics facilitating plant-fungal interactions, these have largely been skewed towards certain fungal clades (e.g. arbuscular mycorrhizal fungi). Several different phyla of fungi have been shown to positively impact plant growth rates, including Mortierellaceae fungi. However, the extent of the plant growth promotion (PGP) phenotype(s), their underlying mechanism(s), and the impact of bacterial endosymbionts on fungal-plant interactions remain poorly understood for Mortierellaceae. In this study, we focused on the symbiosis between soil fungus Linnemannia elongata (Mortierellaceae) and Arabidopsis thaliana (Brassicaceae), as both organisms have high-quality reference genomes and transcriptomes available, and their lifestyles and growth requirements are conducive to research conditions. Further, L. elongata can host bacterial endosymbionts related to Mollicutes and Burkholderia. The role of these endobacteria on facilitating fungal-plant associations, including potentially further promoting plant growth, remains completely unexplored. We measured Arabidopsis aerial growth at early and late life stages, seed production, and used mRNA sequencing to characterize differentially expressed plant genes in response to fungal inoculation with and without bacterial endosymbionts. We found that L. elongata improved aerial plant growth, seed mass and altered the plant transcriptome, including the upregulation of genes involved in plant hormones and "response to oxidative stress", "defense response to bacterium", and "defense response to fungus". Furthermore, the expression of genes in certain phytohormone biosynthetic pathways were found to be modified in plants treated with L. elongata. Notably, the presence of Mollicutes- or Burkholderia-related

**Funding:** GB and PE acknowledges support from the US National Science Foundation DEB 1737898. GB acknowledges support fromthe US Department of Agriculture NIFA MICL02416. The funders had no role in study design, data collection and analysis, decision to publish, or preparation of the manuscript.

**Competing interests:** The authors have declared that no competing interests exist.

endosymbionts in *Linnemannia* did not impact the expression of genes in *Arabidopsis* or overall growth rates. Together, these results indicate that beneficial plant growth promotion and seed mass impacts of *L. elongata* on Arabidopsis are likely driven by plant hormone and defense transcription responses after plant-fungal contact, and that plant phenotypic and transcriptional responses are independent of whether the fungal symbiont is colonized by *Mollicutes* or *Burkholderia*-related endohyphal bacteria.

## Introduction

Microbial promotion of plant growth has great potential to improve agricultural yields and protect plants against pathogens and/or abiotic stresses, while also relieving economic and environmental costs of crop production [1, 2]. Agriculturally important metrics pertaining to plant growth promotion include aerial biomass, root biomass, root architecture, seed number, seed size, and flowering time [3–5]. Early-diverging filamentous fungi in the Mucoromycota are one group of plant beneficial microbes, which have been hypothesized to have assisted plants in the colonization of land [6]. There are three main guilds of plant mutualistic fungi relevant to this study: arbuscular mycorrhizal (AM) fungi, ectomycorrhizal (EM) fungi, and non-mycorrhizal (NM) endophytic fungi. For the purpose of this study, NM root endophytes are defined as fungi that are found inside healthy plant roots but do not make characteristic mycorrhizal structures. Most of these fungi are thought to promote plant growth primarily by providing water and mineral nutrients, and sometimes secondarily by precluding infection by pathogens and/or priming and regulating plant defense responses [7]. However, the signaling mechanisms and fungal symbiotic structures are very distinct between and within these functional guilds, largely because most EM and NM associations represent convergent evolution on a phenotype, rather than a shared evolutionary mechanism of interaction [8].

Mortierellaceae are soil fungi in the subphylum Mortierellomycotina [9]. They are closely related to Glomeromycotina (AMF) and Mucoromycotina, some of which are EM fungi [6, 10, 11]. Plant associations with Mortierellaceae have been recorded since the early 1900s and these fungi are broadly considered NM plant associates [12–14]. Many studies have investigated the impacts of Mortierellaceae fungi on plant growth, however, the extent of the plant growth promotion (PGP) phenotype(s) and the mechanism(s) underlying their association are still not well understood [15–18].

Recent inoculation studies of Mortierellaceae on plant roots show that these fungi elicit a strong PGP phenotype on a broad range of plant hosts [1, 15, 18]. Maize plants inoculated with *Linnemannia elongata* (= *Mortierella elongata*) had increased plant height and dry aerial biomass and analysis of phytohormone levels indicated high levels of abscisic acid and the auxin IAA (indole-3-acetic acid) in response to *L. elongata* [1]. In contrast, *Arabidopsis thaliana* Col-0 (hereafter Arabidopsis) inoculated with *L. hyalina* (= *Mortierella hyalina*) also showed increased total leaf surface area and aerial dry biomass, with reduced levels of abscisic acid and no stimulation of auxin-responsive genes [15]. *Mortierella antarctica* was shown to increase the growth of winter wheat by producing phytohormones IAA and gibberellic acid (GA) and the enzyme ACC (1-aminocyclopropane-1-carboxylate) deaminase, which degrades ACC, a precursor to the phytohormone ethylene [18].

Recent studies have demonstrated Mortierellaceae can harbor endobacterial symbionts [19–22]. However, the impacts of endohyphal bacteria on the PGP phenotype have not been assessed. Although the incidence of endobacteria within isolates of Mortierellaceae is quite low

(<10%), a diversity of bacteria including *Mycoavidus cysteinexigens* and *Mycoplasma*-related endobacteria (MRE) are known to colonize mycelium of diverse species across most of the genera in the family [19–21, 23]. Many species including *L. elongata* can harbor either *Mycoavidus cysteinexigens* or MRE, however, there is generally a single lineage of endobacteria within any particular isolate [19]. Both MRE and *Ca*. Glomeribacter, a *Burkholderia*-related endobacteria (BRE) that is phylogenetically the sister group to *Mycoavidus*, are found in the Glomeromycotina. *Ca*. Glomeribacter has been shown to increase fungal-host biological potential, and is hypothesized to impact plant interactions as a mutualistic partner [24, 25].

In this study, we have focused on interaction between *L. elongata* and Arabidopsis, as both organisms have reference genomes and transcriptomes available. Further, Arabidopsis is small, has a short lifespan, and is ideal for follow-up studies relying on genetic manipulation. We used two isolates of *L. elongata*, NVP64 and NVP80, to better understand mechanisms underlying *L. elongata* symbiosis with plants. These two isolates of *L. elongata* were isolated from the same soil, but were found to be colonized by different endobacteria; NVP64 contains *Mycoavidus cysteinexigens* (BRE) while NVP80 contains MRE, designated as NVP64wt and NVP80wt given that they are the wild-types of these strains. To determine whether either endobacterium has an impact on the plant-fungal symbiosis we generated "cured" isogenic lines of each isolate, NVP64cu and NVP80cu, where the endobacteria were removed through antibiotic passaging. We hypothesized that *L. elongata* would provide a PGP phenotype and that endobacteria would impact this response. We measured PGP of aerial growth at early and late life stages, seed production, and used RNA sequencing to characterize differentially expressed plant genes in response to fungal and endobacteria treatments.

## Materials and methods

### Plant and fungal culturing

**Growth media.** The two fungal strains of *L. elongata* used in this study, NVP64 and NVP80, were isolated from an agricultural soil in Michigan as previously described [9]. were cultured in malt extract broth [MEB: 10 g/L Malt Extract (VWR), 1 g/L Bacto Yeast Extract (Difco, Thomas Scientific; New Jersey, USA)], malt extract agar [MEA: 10 g/L Malt Extract, 1 g/L Bacto Yeast Extract, 10 g/L Bacto Agar (Difco)], and Kaefer Medium [KM: 20 g/L D-Glucose, 2 g/L Peptone, 1 g/L Yeast Extract, 1 g/L Bacto Casamino Acids (Difco), 2 mL/L Fe-EDTA [2.5 g FeSO$_4$*7H$_2$O, 3.36 g Na$_2$EDTA, 500 mL water], 50 mL/L KM Macronutrients [12 g/L NaNO$_3$, 10.4 g/L KCl, 10.4 g/L MgSO$_4$*7H$_2$O, 30.4 g/L KH$_2$PO$_4$], 10 mL/L KM Micronutrients [2.2 g/L ZnSO$_4$*7H$_2$O, 2.2 g/L H$_3$BO$_3$, 0.16 g/L CuSO$_4$*5H$_2$O, 0.5 g/L MnSO$_4$*H$_2$O, 0.16 g/L CoCl$_2$*5H$_2$O, 0.11 g/L (NH$_4$)$_6$Mo$_7$O$_{24}$*4H$_2$O], pH 6.5 with 10 N KOH, and supplemented with Thiamine (1 mg/L) and Biotin (0.5 mg/L) after autoclaving and cooling to 60˚C]. Sterilized seeds were germinated on Murashige & Skoog (MS) medium [1xMS: 4.4 g/L Murashige and Skoog medium (Sigma Aldrich; Missouri, USA), pH 5.7 w/ KOH, and 10 g/L agar (Sigma, product# A1296)]. Plant-fungal experiments were conducted on Plant Nutrient Medium [PNM: 0.5 g/L KNO$_3$, 0.49 g/L MgSO$_4$*7H$_2$O, 0.47 g/L Ca(NO$_3$)$_2$*4H$_2$O, 2.5 mL/L Fe-EDTA, 1 mL/L PNM Micronutrients [4.3 g/L Boric Acid, 2.8 g/L MnCl$_2$*4H$_2$O, 124.8 mg/L CuSO$_4$*5H$_2$O, 287.5 mg/L ZnSO$_4$*7H$_2$O, 48.4 mg/L Na$_2$MoO$_4$*2H$_2$O, 2.4 mg/L CoCl$_2$*6H$_2$O], 10 g/L agar (Sigma, product# A1296), autoclaved and the pH adjusted with 2.5 mL/L 1M H$_2$KPO$_4$ before pouring 22-24mL per 100 mm square plate (with grid)].

To generate a fungal substrate suitable for inoculating potting mix, white millet (Natures Window; Michigan, USA), horticultural perlite (PVP Industries, Inc; Ohio, USA), and pearled barley (International Foodsource; New Jersey, USA) were each soaked overnight in DI water. The millet and barley were each boiled in fresh DI water on a hotplate until the grains began to

break open, then removed from the hotplate and drained of excess water. When prepared, millet and barley expand to about 150% and 300% of the dry volume, respectively. The boiled millet, boiled barley, and perlite were mixed in a 2:1:1 v:v:v ratio. For each treatment, 600 mL of this "millet mix" was placed into a gusseted Unicorn bag (Unicorn Bags, type 10T; Texas, USA) and autoclaved for 45 minutes, allowed to rest overnight under a sterile hood and autoclaved again for 45 minutes.

To generate sterile SureMix-based plant growth substrates, SureMix Perlite (Michigan Growers Products; Michigan, USA) substrate was saturated with deionized water, which was measured and placed into autoclavable bags to ensure the correct volume would be available. A single bag was used for each experimental treatment. The bags of SureMix substrate were autoclaved 45 minutes on a liquid cycle, stored at room temperature for 3–7 days, autoclaved again for 45 minutes on a liquid cycle, cooled to room temperature, and rinsed through with 3 L of sterile MilliQ water (18 MΩ·cm). The autoclaved SureMix was rinsed to remove autoclaving byproducts by flushing with 3 L of sterile MilliQ water on a dish cart covered with a double layer of window screen mesh which had been sterilized with bleach and rinsed with autoclaved MilliQ water.

**Curing fungi of endobacteria.** Replicated lines of *L. elongata* NVP64wt and NVP80wt were cured of their endobacteria by repeated culturing in media containing antibiotics, a protocol adapted from Uehling et al. [20]. Fungi were transferred between MEB and MEA supplemented with 1 g/L Bacto Peptone (Difco), 100 µg/mL ciprofloxacin, 50 µg/mL kanamycin, 50 µg/mL streptomycin, and 50 µg/mL chloramphenicol. Each transfer was performed by transplanting a 1–4 $mm^2$ piece of tissue from the outer edge or surface of the mycelium with a Nichrome inoculating loop and submerging the tissue under the agar surface or broth to maximize contact of the growing hyphae with the antibiotics. Transfers were performed every 3 or 4 days, alternating agar and broth substrate, for 6 transfers.

Following antibiotic curing, tissue from the original and newly-cured lines, as well as the wild-type line, were cultured on antibiotic-free 60 mm MEA plates with an autoclaved cellophane sheet placed atop the agar. After 13 days of incubation, fungal tissue was scraped off the cellophane and DNA extracted using a CTAB-based chloroform extraction protocol (S1 File [26]).

**Arabidopsis seed sterilization & germination.** *Arabidopsis thaliana* Col-0 CS70000 were obtained from the Arabidopsis Biological Resource Center. Seeds were germinated and grown for three generations in a grow room. Bulk seed was collected from the third generation and screened to homogenize seed size with 350 µm and 250 µm sieves (VWR, Pennsylvania, USA), retaining the middle fraction.

Arabidopsis seeds were divided from the screened stock into 1.5 mL Eppendorf tubes using a 200 seed spoon, with up to 1200 seeds per tube. Seeds were surface sterilized by washing in 800 µL 70% Ethanol for 20 seconds, discarding the ethanol, and then washing in 400 µL 20% bleach (Clorox Performance, 8.3% Sodium Hypochlorite, Clorox, California, USA) for 30 seconds. Seeds were then rinsed of bleach three times by quenching with 1 mL sterile water and discarding the liquid. Seeds were then resuspended in 500 µL sterile water prior to planting.

Surface sterilized seeds were plated on 1xMS using a p1000 and sterile water, 16 seeds per plate in rows of 3, 4, 5, and 4, with about 1cm between seeds and rows (**S1A Fig**). We germinated at least 5 times as many seeds as were needed for the experiment to allow greater control of seedling size.

Germination 1xMS plates were cold stratified for 2 days in the dark at 4˚C to synchronize germination, then allowed to germinate and grow for 5 or 10 days, depending on the experiment, in a Percival I-36LLVL growth chamber at 103–118 µmol/$m^2$·s PAR with 16 hr day & 8 hr night, 22˚C, ambient humidity. Light levels were measured using an LI-250A light meter (LI-COR, Nebraska, USA).

## Potting mix experiments

**Grain-based inoculum.** Each fungal strain was grown in 250 mL Erlenmeyer flasks with 75 mL of MEB for 2 weeks. Colonized medium was poured out into an autoclaved beaker and the mycelium collected with sterile tweezers, coarsely chopped in a sterile petri dish, and added to sterile millet mix bags. The bags were lightly mixed, sealed in two places with an impulse sealer, and the fungi allowed to colonize the spawn for two weeks.

**Arabidopsis growth conditions.** Five days after germination, Arabidopsis seedlings were transplanted from 1xMS plates to plug trays of autoclaved and rinsed SureMix and moved to a Bio Chambers AC-40 growth chamber with 16 hr day, 8 hr night, 22˚C, ambient humidity. Seedlings were grown in plugs for 11 days (16 days after germination). The soil plugs and seedling roots were treated with Zerotol 2.0 (BioSafe Systems, Connecticut, USA), an algaecide, bactericide, and fungicide containing hydrogen peroxide and peroxyacetic acid. The Zerotol was applied as a soil drench for 15 minutes, rinsed three times with distilled water, and transplanted into 4 in$^3$ pots with SureMix mixed with the appropriate millet treatment mixed at 3% by volume (except the NoMillet treatment, which was 100% SureMix). Each treatment was contained in a separate waterproof tray arranged with three rows of six pots. Using seventeen pots per treatment left an empty spot for watering. Four days after transplanting, seedlings were treated with 2 L of Peters 20-20-20 fertilizer mixed at 1/8th strength (0.1 tsp/L) in MilliQ water. Thereafter, plants were watered with MilliQ water as needed.

**Above ground biomass.** At 34 days after transplanting and inoculation (50 days after germination), all treatments were observed to have ripening siliques, necessitating harvesting to avoid excessive loss of seed biomass during plant handling. Twelve plants per treatment were harvested by cutting the roots at the potting mix line and trimming and/or folding the aerial parts into tared envelopes (Top Flight no.10 Security Envelope, Strip & Seal). Fresh weight was recorded immediately after harvesting was complete. Plants were dried at room temperature (20–22˚C) for 2 weeks and re-weighed for the dry biomass. All envelope and plant biomass measurements were taken on a Mettler Toledo PG2002-S scale.

**Seed collection.** Five plants were randomly selected for seed collection. ARACON tubes (Arasystem, Belgium) were installed over the rosette. When the remaining plants were harvested for biomass, these five plants were moved to a drying room for two weeks. Dry plant material was collected and stored in wax paper bags until processing. Seeds were isolated from plant material by manually massaging the bags to release seeds, filtering through a Rösle Stainless Steel Fine Mesh Tea Strainer (Wire Handle, 3.2-inch, model# 95158) to remove large plant debris, repeatedly passing over copier paper, and picking out remnant plant matter with tweezers. Cleaned seeds were collected in tared 2 mL Eppendorf tubes and weighed on a Mettler Toledo AB104-S/FACT scale. To determine average seed mass, approximately 14 mg of seeds per sample were weighed on an ultrasensitive balance, adhered to a piece of white paper using a glue stick, covered by clear packing tape, scanned, and counted by image analysis in ImageJ following protocols optimized by Dr. Mathew Greishop's lab, based on the work of Mark Ledebuhr (S1 File and **S2 Fig**).

**Statistical analysis.** Since the data were non-normal, we performed Wilcox ranked sum tests and adjusted p-values for multiple comparisons using the Holm method. Between NVP64cu v. NVP64wt, NVP80cu v. NVP80wt, and NoMillet v. Uninoculated, we used two-tailed tests. Between each fungal treatment and the Uninoculated, we performed one-tailed tests with the alternative hypothesis being "less" or "greater" as appropriate. Data analysis and visualization was conducted in R using the ggpubr and ggsignif packages [27, 28]. Datasets and code are available at https://github.com/natalie-vandepol/Arabidopsis-L.elongata-PGP.

## Agar-based experiments

**Transplanting & inoculation.** We based the design of these experiments on the methodology used by Johnson et al. [15]. Arabidopsis seeds were surface sterilized and germinated as described previously. Ten days after germination, seedlings were categorized into three approximate seedling size "categories": too small, too big, and average. Three "average" seedlings were transplanted to each PNM plate such that the cotyledons aligned with the top line of the plate grid and the roots were not covered by the grid pattern (**S1B Fig**). Each plate was numbered as it was populated with seedlings so that plates could be assigned to treatments serially (e.g., 1-A, 2-B, 3-C, 4-A, 5-B, 6-C, 7-A, etc.), to homogenize variation and bias in seedling size throughout the transplant procedure. Plates were inoculated by transferring two 5 mm x 5 mm squares of Kaefer Medium, sterile or colonized by the appropriate fungal culture, between the three seedlings. The number of biological replicates per treatment varied by experiment as follows: 27 for the bolting assay, 12 for the media panel, and 27 for the endobacteria panel.

**Root length.** After transplanting and inoculation, seedlings and fungi were left undisturbed overnight to allow them to adhere to the media and minimize the likelihood of movement during handling. The following day (1 day post inoculation), plates were imaged on an Epson scanner at 1200 dots per inch using Home mode and default settings (**S1b Fig**). Images were processed in ImageJ v.1.52p, using the 13 mm grid on the plates as a scale, the freehand line tool to trace the roots, and the measuring tool to determine starting root length of each seedling.

**Growth chamber.** Light levels were measured with a LI-250A light meter (LI-COR) at 9 different points on each of the four shelves in the growth chamber (**S1 Table**). To homogenize variability in environmental conditions across treatments, plates were distributed between light level regions and the lower three shelves as evenly as possible and their location in the chamber recorded. Each of the shelves accommodate 3 rows of 15 plates, with 5 plates assigned to each of the 9 zones on the shelf (**S3 Fig**).

**Bolting panel.** To determine whether bolting time was affected by fungal colonization, PNM plates with 10 day old Arabidopsis seedlings were inoculated and monitored daily for evidence of bolting, which was defined as visible elongation of the emerging inflorescence away from the rosette (**S4 Fig**). As each plant bolted, the date was noted on the plate.

**Harvesting aerial plant material.** At 12 DPI (22 days old), the aerial portion of each plant was cut away from the roots and placed into a folded "envelope" made from weighing paper and dried in a 65°C drying oven for 48 hours. The envelopes of dried plants were stored in empty tip boxes and double bagged with Ziplock bags to prevent reabsorption of atmospheric water before weighing. Dry plants were weighed on a DeltaRange XP26 ultrasensitive balance (Mettler Toledo; Ohio, USA).

**Statistical analysis.** We conducted statistical analyses in R v.3.6.0 using the tidyverse v1.3.0, lme4 v1.1–21, lmerTest v3.1–1, car v3.0–6 packages [29–33]. Bolting data were visualized as boxplots and visibly non-normal. We used the Kruskal-Wallis test to examine differences in bolting age between treatments [34].

Aerial dry weight data were visualized as boxplots and assessed as approximately normal and homoskedastic. We used analysis of variance (ANOVA) and linear models to examine differences in dry weight within each experimental dataset to determine the effects of environmental factors tested by each experiment. Based on the results of these tests, we constructed a linear mixed model of the combined dry weight data from the two agar experiments, specifying treatment and seedling root length as fixed effects and experiment (Media Panel & Cured Panel) and plate (to account for three plants measured per plate) as grouping factors:

$$DryWeight \sim Treatment + RootLength + (1 \mid Experiment : Plate)$$

We used the emmeans v1.4.4 package to perform pairwise comparisons of the model estimates for each treatment [35]. The estimated marginal mean, confidence interval, and significance groups were extracted for graphical summarization.

**Root microscopy.** Seedlings were grown on MS plates (see above) for five days post-stratification, then transferred to PNM plates, with four seedlings per plate, approximately 1 cm from the plate edge. Two ~5 mm x 10 mm blocks of MEA colonized with mycelium of L. elongata NVP64cu or NVP80cu were placed on the centerline of the plate, spaced between plants 1 and 2 then 3 and 4. The plates were sealed with parafilm and arranged vertically in a Percival I-36LLVL growth chamber at 75 μmol/m$^2$·s PAR with 16 hr day & 8 hr night, 22˚C, 60% relative humidity. Roots were sampled at 13 DPI (18 days old) and again at 23–25 DPI (28–30 days old).

Roots were cut from shoots using a scalpel, then forceps were used to transfer roots to 200 μL of stain solution or 1x PBS, pH 7.2, in a 500 μL conical tube. The stain solution was composed of 25 μg/mL WGA-640R (Biotium, California, USA) and 1 mg/mL calcofluor white M2R, in 1x PBS, pH 7.2. Vacuum was applied to the roots in liquid with a vacuum pump, three times for 30 s each, releasing the pressure after each time. The roots in stain or buffer were incubated at room temperature for 30 min on a table shaker at 60 rpm.

Roots were removed from the stain solution and placed on glass slides, then coverslips were added. Roots were imaged using an Olympus Fluoview FV10i confocal laser scanning microscope. The unstained roots were viewed first and used to calibrate sensitivity values. The WGA-640R channel was viewed with $\lambda_{ex}$ ~ 642 nm, $\lambda_{em}$ ~ 661 nm and the calcofluor white M2R channel was viewed with $\lambda_{ex}$ ~ 352 nm, $\lambda_{em}$ ~ 455 nm. The micrographs were processed, recolored, and transformed in ImageJ v1.53h [36] and 3D Slicer v4.11.20210226 [37].

## RNA sequencing & differential gene expression

**Root harvesting & storage.** The root material for the RNAseq experiment was collected from the plants generated in Agar experiments (**Fig 1**, see Agar-based experiments: Harvesting aerial plant material Methods section). Before collecting the aerial parts of the plants for biomass assays, five plates were selected from each treatment on the basis of similar light levels within the chamber, three of which were selected as triplicate biological replicates for RNA sequencing based on having aerial dry weights closest to the mean for that treatment. For each selected plate, two RNAse-zap treated, DEPC water rinsed, autoclaved steel beads were placed in one RNAse-free 1.5 mL Eppendorf tube, handled with gloves treated with RNAse-zap. Eppendorf tubes were placed in an autoclavable tube box, open and upright, the box wrapped in foil and autoclaved for 25 minutes on a dry cycle. After autoclaving, wearing RNAse-zap treated gloves, the Eppendorf tubes were carefully removed from the box, closed, and labeled with the numbers of the plates from which the roots were to be collected.

Ten days after germination, *Arabidopsis thaliana* seedlings were transplanted from 1xMS germination plates to these PNM plates and inoculated with small blocks of Kaefer Medium, either colonized by fungi (**left**) or sterile (**right**). The Arabidopsis (and fungi, when applicable) grew on PMN plates for 12 days, at which point these pictures were captured and the plants harvested for aerial biomass assays.

During harvest, each selected plate was removed individually from the chamber, opened, and the roots collected with forceps and a scalpel. The roots were immediately placed in a cold Eppendorf tube and flash frozen in liquid nitrogen. The time between removing the plate from its place in the chamber to freezing the Eppendorf tube and roots did not exceed 30 seconds. The forceps and scalpel were soaked in 10% bleach between samples and excess liquid wicked off by a paper towel before contacting the roots. The Eppendorf tubes of root samples were stored at -80˚C prior to extracting RNA.

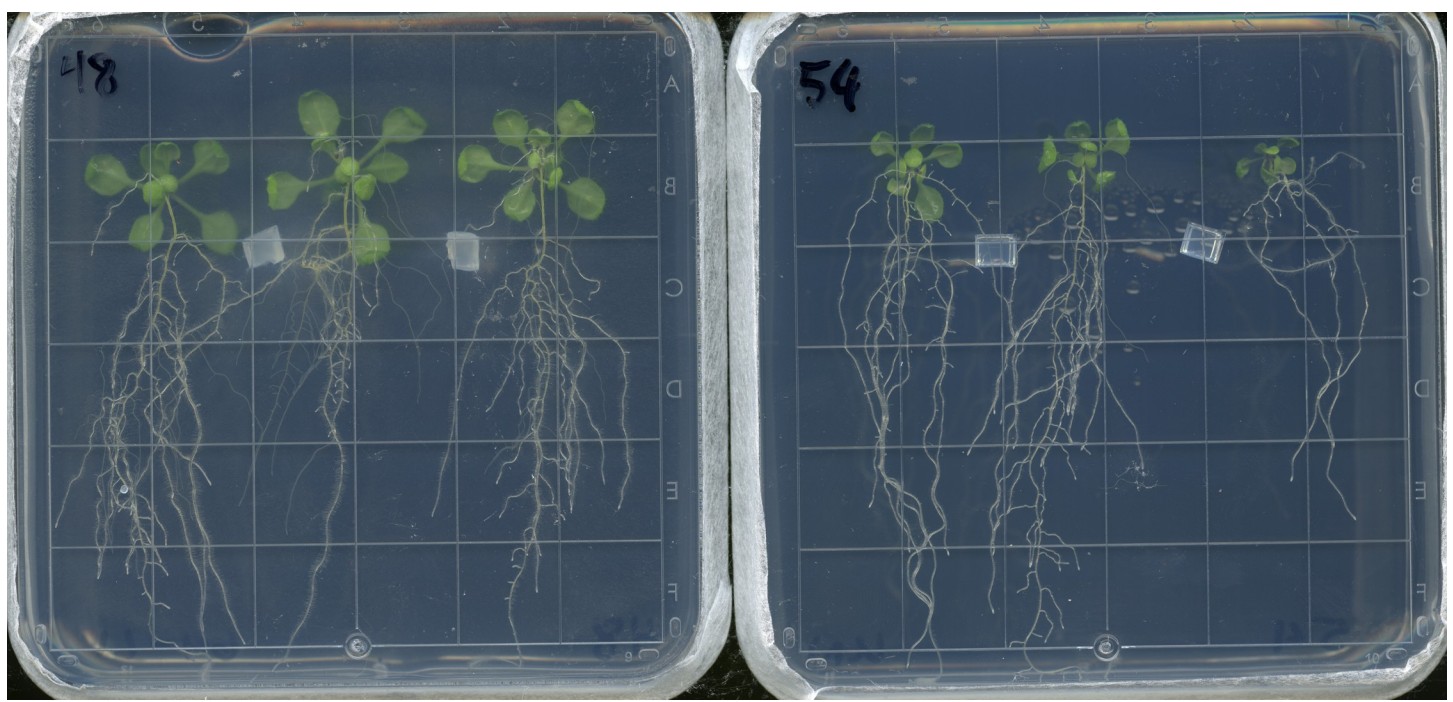

**Fig 1. Arabidopsis plants at the time of harvest for aerial biomass assay.**

**RNA extraction.** Tissue was homogenized by three 30 second bursts at 30Hz in a Tissue-Lyzer II (Qiagen; Germany), with 30 second rests in liquid nitrogen between each burst. RNA was extracted using a Qiagen RNEasy Plant Mini Kit, employing 450 μL Buffer RLT lysis buffer (with 10 μL β-ME per 1 mL Buffer RLT), an on-column DNAse digest (RNase-Free DNase Set, Qiagen), and eluting twice with 50 μL RNAse free water. A 5 μL aliquot was set aside to perform an initial quantification using a NanoDrop. Samples with less than 75 μg/mL were concentrated by ethanol precipitation as described below. RNA was quantified and quality checked using BioAnalyzer (MSU RTSF). All RNA samples had RIN scores >9.0.

**RNA ethanol precipitation.** Low concentration RNA extractions were amended with 10 μL 3 M Sodium Acetate and then 300 μL ice cold 100% ethanol, vortexed briefly to mix, and precipitated at -20˚C overnight. RNA was pelleted by centrifuging for 30 min at full speed at 4˚C. The RNA pellet was washed with 200 μL ice cold 70% EtOH and centrifuged for 10 min at full speed at 4˚C. The supernatant was discarded and the pellet air-dried for 15 min on the bench before being resuspended in 30–50 μL RNAse-free water. A 5 μL aliquot was taken for quantity and quality analysis and the remainder stored at -80˚C.

**Library preparation & sequencing.** Libraries were prepared using the Illumina TruSeq Stranded mRNA Library Preparation Kit with the IDT for Illumina Unique Dual Index adapters following manufacturer's recommendations. Completed libraries were QC'd and quantified using a combination of Qubit dsDNA HS and Agilent 4200 TapeStation High Sensitivity DNA 1000 assays. The libraries were pooled in equimolar amounts and the pool quantified using the Kapa Biosystems Illumina Library Quantification qPCR kit. This pool was loaded onto an Illumina NextSeq 500/550 High Output flow cell (v2.5) and sequencing performed in a 1x75 bp single read format using a NextSeq 500/550 High Output 75 cycle reagent kit (v2.5). Base calling was done by Illumina Real Time Analysis (RTA) v2.4.11 and output of RTA was demultiplexed and converted to FastQ format with Illumina Bcl2fastq v2.19.1. Sequence data have been accessioned in NCBI's SRA under the BioProject PRJNA704083.

**qPCR.** Primer sets for qPCR were designed using 16S rRNA gene sequences of *L. elongata* NVP64 and NVP80 endobacteria with the IDT PrimerQuest® Tool for 2 primers and intercalating dye (**S2 Table**). Primer sets were verified using wild-type DNA samples, for which a standard curve was created with dilutions from 100 to $10^{-4}$ and efficiencies were within 90–110%. Absolute copy number calibration was not performed because only presence/absence validation was required. cDNA was synthesized for qPCR quantification using the LunaScript RT SuperMix Kit (New England Biolabs; Massachusetts, USA). qPCR reactions were composed of 7.5 μL Power SYBR Green PCR Master Mix (ThermoFisher Scientific; Massachusetts, USA), 5.5 μL nuclease-free water, 0.25 μL each primer, and 1.5 μL of undiluted template. The reaction cycle was 95˚C for 10 min, followed by 40 cycles of 95˚C for 15 sec and 60˚C for 1 min with a fluorescence measurement. A melting curve was performed following amplification: 95˚C for 15 sec and 60˚C for 15 sec, then a 20 min ramp up to 95˚C, followed by 95˚C for 15 sec. At least two reactions were performed per sample and primer combination.

**Sequence analysis.** Raw, demultiplexed reads were quality trimmed and filtered using Trimmomatic v.0.38 [38]. A combined reference transcriptome was constructed from the Arabidopsis Thaliana Reference Transcript Dataset 2 (AtRTD2_19April2016.fa, accessed 10/21/2019) and *Linnemannia elongata* AG77 (Morel2_GeneCatalog_transcripts_ 20151120.nt.fasta.gz, project 1006314, accessed 10/21/2019) [16, 20]. This combined reference transcriptome was indexed in Salmon v0.11.3 and used to quasi-map the trimmed reads to transcripts [39].

**Differential gene expression analysis.** A transcript-to-gene (tx2gene) table was constructed in R v.3.6.0 for Arabidopsis gene annotations (AtRTD2_19April2016.gtf.txt, accessed on 01/12/2020). Fungal reads were extremely rare in the dataset, thus, analyses focused solely on plant transcriptional responses [29, 40]. Salmon quants.sf files were imported into R using tximport (type = salmon) [41]. Differential gene expression analysis was carried out with both the EdgeR package and the DESeq2 package in R [42–44]. Gene expression was computed for each treatment across the three biological replicates, with the control treatment specified as the reference level in the experimental design matrix. Differentially expressed genes were identified by contrasting each fungal treatment against the control. In DESeq2, gene lists from each comparison were filtered by an adjusted p-value of 0.05 and an absolute value of log2 fold change (LFC) cutoff of 0.585, which corresponds to a fold change in expression of 1.5. We generated volcano plots of these pairwise comparisons using the EnhancedVolcano package in R [45]. In EdgeR, the gene list encompassed all four fungal treatments with a single p-value for each gene, so the EdgeR gene list was filtered by overall p-value and whether at least one fungal treatment LFC meeting the LFC cutoff [42, 43]. The DESeq2 gene list was filtered to include only genes also present in the EdgeR gene list. Since DESeq2 provided p-values for each comparison, we used the log2-fold change and adjusted p-value of the DESeq2 analyses to compose our final DEG table. Gene ontology was assigned by referencing TAIR and UniProtKB annotation databases and synthesizing the most detailed and supported annotations [46, 47].

**Functional enrichment.** We generated a list of differentially expressed genes in response to at least a single fungal treatment. The list of up and down regulated genes were separately searched for functional enrichment using the clusterprofiler package in R. Code to reproduce the GO enrichment is publically available: https://github.com/Aeyocca/00_Collab/tree/main/plant_fungal_interactions (last accessed 11-03-2021) [48].

## Results

### Potting mix experiments

**Linnemannia elongata *increased mature Arabidopsis aerial dry biomass*.** All fungal treatments had significantly higher aerial dry biomass than the uninoculated millet control.

Aerial dry biomass of full-grown Arabidopsis plants was not significantly different between NVP64cu and NVP64wt or between NVP80cu and NVP80wt (**Fig 2**). Millet has previously been used as a fungal substrate for inoculating soil in plant-fungal symbiosis research [16, 49]. A NoMillet control was initially included to test the assumption that the millet-based inoculum had no impact on the plants. However, the NoMillet controls had the highest aerial dry biomass of any experimental treatment, indicating that the millet carrier negatively impacts plant health. Thus, results of the potting mix experiments may be interpreted in terms of stress mitigation. In this case, the NoMillet control presents a baseline and the Uninoculated control is an unmitigated stress imposed by the millet grain. The fungal treatments generally fell between these two treatments, indicating partial mitigation of the stress imposed by the grain-based inoculum. Given that the exact nature of the stress imposed by the grain-based inoculum is unknown, we focused our analyses of these data on fungal treatments v. uninoculated control and relied on pure culture agar plate methods for subsequent experiments.

*Arabidopsis thaliana* was grown to maturity and the aerial biomass harvested and dried. Treatments refer to the composition of the potting mix. The untreated control (NoMillet)

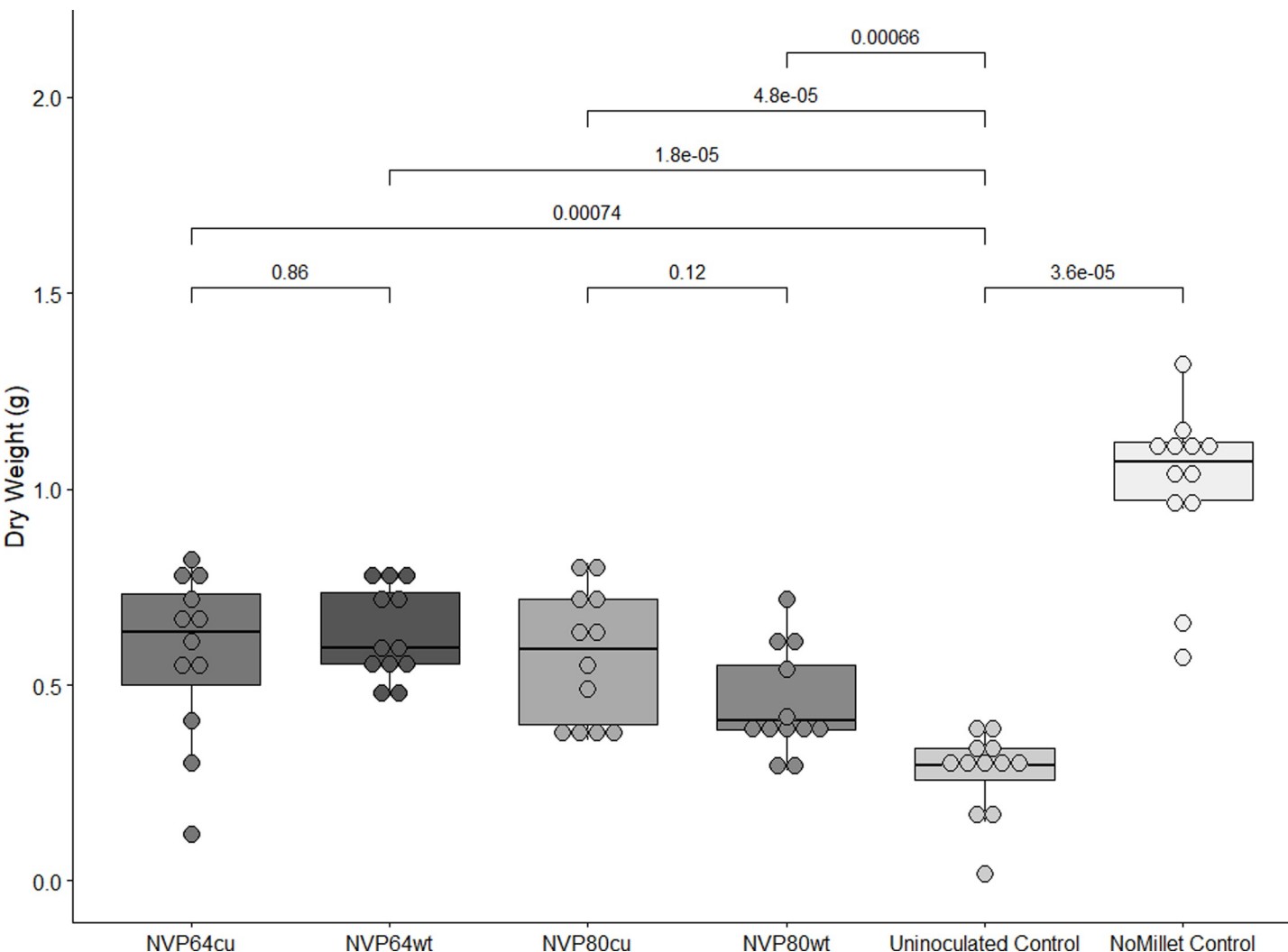

**Fig 2. Aerial dry biomass of Arabidopsis plants grown in sterile potting mix.**

contrasted treatments where the sterile potting soil was mixed 97:3 v:v with sterile millet mix (Uninoculated), or millet mix inoculated with one of four *Linnemannia elongata* strains (NVP64cu, NVP64wt, NVP80cu, or NVP80wt). Colors correspond to treatment, horizontal brackets and numbers indicate pairwise Wilcox ranked sum tests and the resulting p-value. N = 12 for all treatments. Between NVP64cu v. NVP64wt, NVP80cu v. NVP80wt, and NoMillet v. Uninoculated, we used two-tailed tests. Between each fungal treatment and the Uninoculated, we performed one-tailed tests with the alternative hypothesis "greater".

**Linnemannia elongata *impacted Arabidopsis seed production.*** As with the aerial biomass, the total seed mass of NVP80cu and NVP80wt were significantly higher than the Uninoculated control (**Fig 3A**). No significant differences in total or average seed mass were found between the isogenic isolate pairs, i.e. NVP64wt vs. NVP64cu and NVP80wt vs. NVP80cu (**Fig 3**). Unlike total seed mass, the average seed mass of the Uninoculated control was slightly higher than NVP80wt and NVP64cu, but not significantly different from the NoMillet control (**Fig 3B**). The total seed mass in the NoMillet control was significantly higher than that of the Uninoculated control.

Given the potential that sufficient seeds in the fungal treatments could be smaller due to incomplete development, rather than total reduction in seed size, we set out to determine whether this might be visible in violin plots of individual seed pixel areas from the image analysis. This would be represented by a bimodal violin with peaks representing immature and mature seeds. No strong bimodality could be seen in replicates or treatments (**S5 Fig**).

## Agar experiments

**Linnemannia elongata *did not impact the timing of Arabidopsis bolting.*** The Kruskal-Wallis Test was conducted to examine the age at which plants bolted according to treatment. With 27 plates per treatment and 3 plants per plate, no significant differences in bolting time ($H = 4.92$, $p = 0.296$, df = 4) were found between the five treatments. The mean age at which

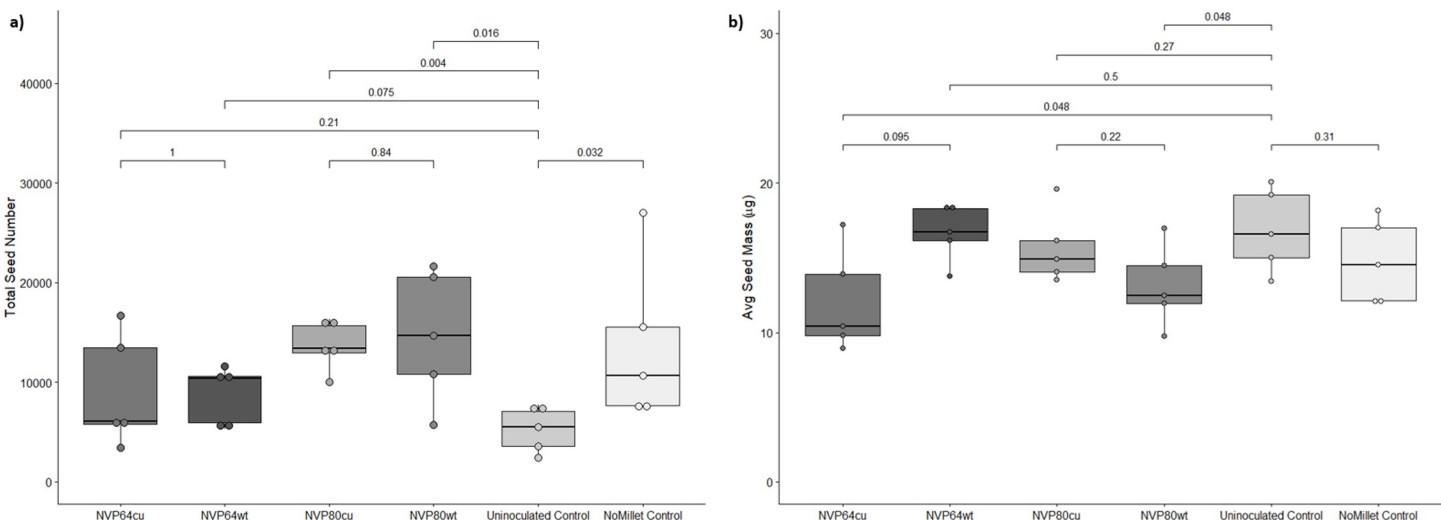

**Fig 3. Total and average Arabidopsis seed mass collected in potting mix experiments.** *Arabidopsis thaliana* was grown to maturity and the seeds collected by Aracon tubes. Treatments refer to the composition of the substrate in which Arabidopsis plants were grown. The NoMillet Control was autoclaved SureMix. All other treatments were autoclaved SureMix substrate mixed 97:3 v:v with sterile grain-based inoculum (Uninoculated), or grain-based inoculum colonized by one of four *Linnemannia elongata* strains (NVP64cu, NVP64wt, NVP80cu, or NVP80wt). N = 5 for all treatments. Colors correspond to treatment, horizontal brackets and numbers indicate pairwise Wilcox ranked sum tests and the resulting p-value. Between NVP64cu v. NVP64wt, NVP80cu v. NVP80wt, and NoMillet v. Uninoculated, we used two-tailed tests. Between each fungal treatment and the Uninoculated, we performed one-tailed tests with the alternative hypothesis "greater" for total seed mass & "less" for average seed mass. **a)** Total seed mass collected. **b)** Average seed mass was determined by weighing and then counting a subset of seeds taken from the total seed mass.

an inflorescence could first be seen to elongate from the rosette was 22 days old, which was 12 days post transplanting and inoculation (DPI). Therefore, we harvested all further agar experiments at 12 DPI to prevent bolting from affecting dry weight data, which differed from the 7 day co-cultivation time used by Johnson et al. [15] (**Fig 1**).

**Linnemannia elongata *increased young Arabidopsis aerial dry biomass..*** We expected that several environmental factors could potentially impact our observation of how Arabidopsis responds to *L. elongata*. These included the (1) starting size of the plant; (2) local light level, (3) medium on which the fungus was cultured, and (4) process by which the fungi were cured of their endobacteria. We determined that there was no statistically significant correlation between light level and harvested plant dry weight in any of the treatments (**S3 Table**). We performed linear modeling of the dry weights as a function of medium, treatment, and interaction between those, and determined there were no significant differences in harvested plant dry weight based on media ($F_{1,110} = 0.966$, $p = 0.328$; **S4 Table**) and no significant interaction between medium and treatment ($F_{4,110} = 0.331$, $p = 0.857$). Analysis of variance found no statistically significant differences in mean harvested plant dry weight, between three independently generated cured lines (L0, L1, and L2) of *L. elongata*, for both NVP64 ($F_{2,42} = 0.443$ $p = 0.645$) and NVP80 ($F_{2,42} = 1.966$, $p = 0.153$), indicating that differences between wild-type (wt) and cured (cu) strains are likely due to the presence/absence of endobacteria, rather than accumulated mutations from the antibiotic passaging protocol. Analysis of variance in seedling root length showed that the mean seedling root length was consistent between treatments of each experiment ($F_{4,755} = 0.953$, $p = 0.433$), but differed significantly between experiments ($F_{1,755} = 267.3$, $p = 2e\text{-}16$), with no significant interaction effect ($F_{4,755} = 0.541$, $p = 0.706$). Preliminary linear model analysis showed a significant positive correlation between seedling root length and harvested plant dry weight, with no significant differences between the slope of this correlation across experiments or treatments (**S5 Table**). We fit a linear mixed model of combined aerial dry weight data from both experiments as a function of treatment and seedling root length. Results of this model can be seen in **Table 1**. The estimated marginal means of aerial dry weight was significantly higher in all four fungal treatments compared to the control, but there were no significant differences between fungal treatments (**Fig 4**).

The estimated marginal mean of *Arabidopsis thaliana* aerial dry weight, modeled as a function of starting root length and treatment with nested effects for experiment iteration (3 independent iterations) and agar plate (3 plants per plate). Treatments included the uninoculated control and four strains of *Linnemannia elongata*. N = 39 plates for Control, NVP64wt, and NVP80wt, nN = 69 plates for NVP64cu and NVP80cu. The degrees of freedom for each comparison were approximated using the kenward-roger method and the p-values adjusted for multiple comparisons using the Tukey method for comparing a family of 5 estimates. Letters indicate significantly different groups with an alpha value of 0.05. Exact values can be found in **Table 1**.

***All* Linnemannia elongata *strains colonize Arabidopsis roots evenly.*** We used the cycle number at which the fluorescent signal of the qPCR probe exceeded the threshold level to calculate the ratio of *L. elongata* RNA to Arabidopsis RNA in each reaction. This ratio represents the degree of fungal colonization of plant roots. There were no significant differences in this ratio between any of the fungal treatments ($p > 0.1$) and each lineage of endobacteria was detected only in the wild-type strains (**S6 Table** and **S6 Fig**).

We visually assessed the ability of *L. elongata* NVP64cu and NVP80cu to grow on and into root tissue, and the localization of hyphae within the roots for plants grown on agar. At 13 DPI *L. elongata* had colonized the root rhizosphere, but no internal hyphae were observed for NVP80cu. However, by 23 DPI we observed NVP80cu hyphae within epidermal root cells and root hair cells, with clearly visible plant cell walls bounding the hyphae on all sides (**Fig 5A–**

**Table 1. Linear mixed modeling of Arabidopsis aerial dry weight.**

| Fixed Effects | | | | | |
|---|---|---|---|---|---|
| | Estimate | Std.Error | df | t-value | *p* |
| (Intercept) | 0.581 | 0.098 | 8.502 | 5.92 | 2.7E-04 |
| treatment = NVP64cu | 0.601 | 0.068 | 230.9 | 8.79 | 3E-16 |
| treatment = NVP64wt | 0.565 | 0.076 | 232.6 | 7.41 | 2E-12 |
| treatment = NVP80cu | 0.650 | 0.068 | 230.8 | 9.52 | <2E-16 |
| treatment = NVP80wt | 0.681 | 0.076 | 231.9 | 8.93 | <2E-16 |
| Root Length | 0.122 | 0.008 | 514.1 | 16.06 | <2E-16 |
| Random effects | | | | | |
| | Name | Variance | Std.Dev. | # of Groups | |
| Plate | (Intercept) | 0.074 | 0.273 | 255 | |
| Experiment | (Intercept) | 0.005 | 0.072 | 2 | |
| Residual | 0.117 | 0.342 | - | - | |
| EMM Pairwise Comparisons | | | | | |
| Contrast | estimate | SE | df | t.ratio | *p* |
| Control—NVP64cu | -0.6005 | 0.069 | 250 | -8.7 | < .0001 |
| Control—NVP64wt | -0.5654 | 0.0763 | 249 | -7.41 | < .0001 |
| Control—NVP80cu | -0.6498 | 0.0689 | 249 | -9.43 | < .0001 |
| Control—NVP80wt | -0.6807 | 0.0762 | 248 | -8.93 | < .0001 |
| NVP64cu—NVP64wt | 0.0351 | 0.0689 | 250 | 0.509 | 0.986 |
| NVP64cu—NVP80cu | -0.0494 | 0.0573 | 249 | -0.86 | 0.911 |
| NVP64cu—NVP80wt | -0.0803 | 0.069 | 250 | -1.16 | 0.772 |
| NVP64wt—NVP80cu | -0.0844 | 0.0689 | 250 | -1.23 | 0.736 |
| NVP64wt—NVP80wt | -0.1153 | 0.0763 | 249 | -1.51 | 0.556 |
| NVP80cu—NVP80wt | -0.0309 | 0.0689 | 249 | -0.45 | 0.992 |

To account for having measurements for three plants per agar plate and two independent repetitions of the agar-based interaction experiment, experimental round and plate were treated as random/grouping effects. The starting root length and experimental treatment were fixed effects, where the uninoculated control treatment was estimated as the intercept.

**5F**). Similarly, we visualized NVP64cu growing to high density within epidermal cells at 25 DPI, and the mass of hyphae bounded by the plant cell wall (**Fig 5G–5I**).

## Differential gene expression

**Molecular results.** We generated a total of 521.2 million sequence reads (39.1 Gb) at an average of 34.7 million (30.5–37.8M) sequence reads per sample, with an average of 97.64% (97.22–97.85%) mapping rate to the combined reference transcriptome. Of the mapped reads, an average of 99.82% (98.64–99.99%) mapped to plant transcripts (**S7 Table**). Thus, analyses were focused on plant responses to experimental treatments.

**Arabidopsis differentially expressed genes in response to *Linnemannia elongata*.** We conducted initial RNAseq data exploration in DESeq2 to confirm consistent gene expression profiles between biological replicates of each condition. Our principal component analysis showed that all four fungal treatments clustered together and away from the control (**S7 Fig**). However, there was no observed clustering by isogenic strain (NVP64 or NVP80; PERMA-NOVA p-value = 0.09) or by cured/wild-type (PERMANOVA p-value = 0.051). Indeed, NVP64cu and NVP80wt seem to be the most similar.

DESeq2 provided p-values for each comparison, and we used the log2-fold change (LFC) and adjusted p-value of the DESeq2 analyses to filter the expression patterns in the final DEG

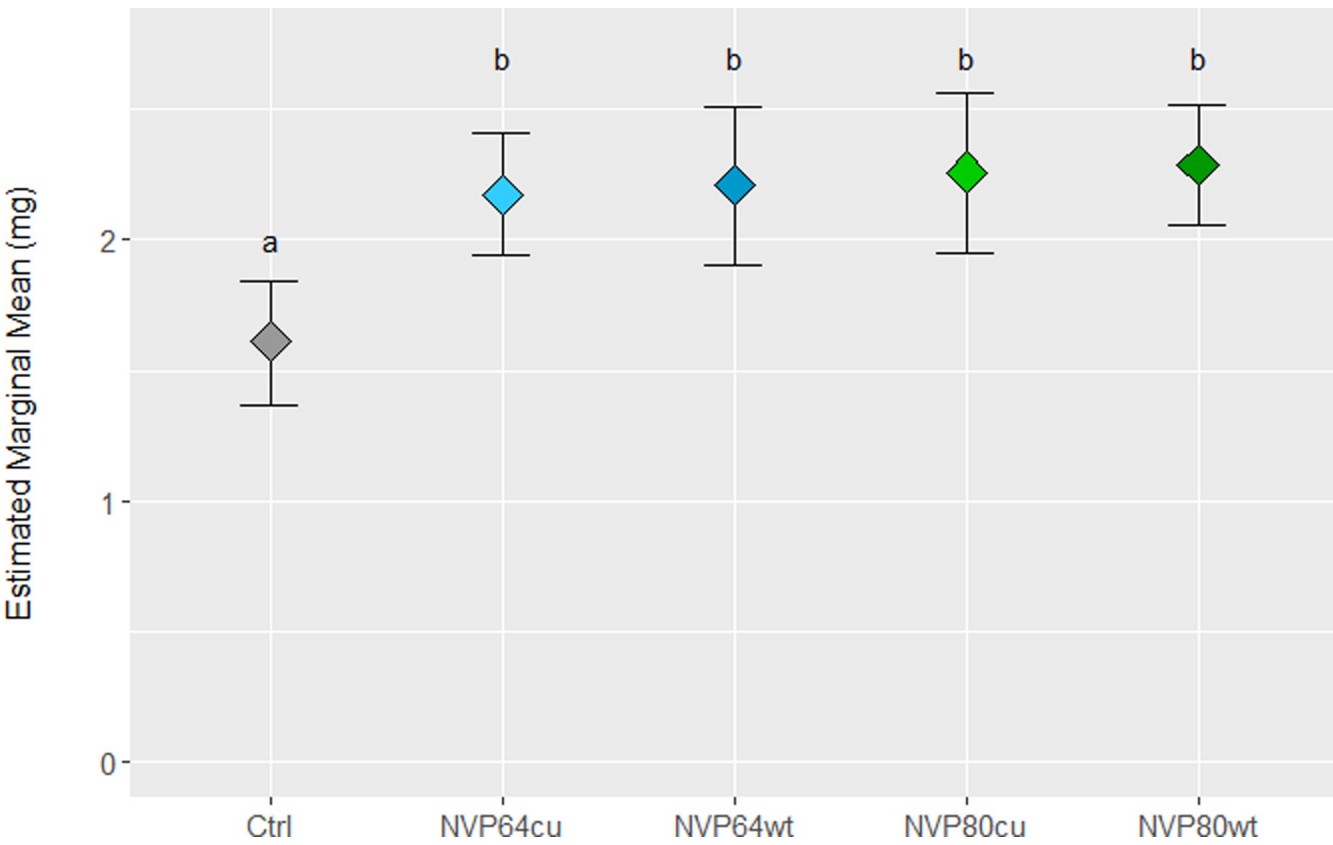

**Fig 4.** *Linnemannia elongata* **colonization of Arabidopsis increased aerial dry weight in agar-based interaction experiments**.

list. DESeq2 identified a total of 465 genes that were differentially expressed and met LFC and adjusted p-value cutoffs in at least one of the four fungal treatments as compared to the control. Of these, there were 301 differentially expressed genes (DEGs) in NVP64cu v. Control, 135 in NVP64wt v. Control, 142 in NVP80cu v. Control, and 213 in NVP80wt v. Control (**S8 Fig**). EdgeR identified 679 genes as being differentially expressed at a collective adjusted p-value threshold, with at least one sample meeting the LFC cutoff. There were 376 DEGs in NVP64cu v. Control, 240 in NVP64wt v. Control, 282 in NVP80cu v. Control, and 319 in NVP80wt v. Control. We identified 385 DEGs present in both the EdgeR and DESeq2 differentially expressed genes results (**S8 Table** and **Fig 6**).

Thirty-four plant genes were differentially expressed when inoculated with all of the four fungal treatments as compared to the uninoculated control, 55 in three fungal treatments, 114 in at least two fungal treatments, and 182 in only one fungal treatment (**S8 Table**). Differentially expressed genes responded in the same direction to treatments, with only one exception (**S8 Table**). **Table 2** highlights a subset of twenty-five DEGs having particularly interesting gene function and consistent significance across multiple fungal treatments.

**Gene ontology enrichment of differentially expressed genes.** Next, we ran Gene Ontology enrichment analysis on the differentially expressed genes (DEGs) that responded to at least a single fungal inoculation. There were 172 upregulated and 212 downregulated genes. Several biological processes were enriched among these DEGs (**Fig 7** and **Table 2**). In response to fungal treatment, upregulated genes were strongly enriched for "response to oxidative stress" (GO:0006979), "defense response to bacterium" (GO:0042742), and notably "defense

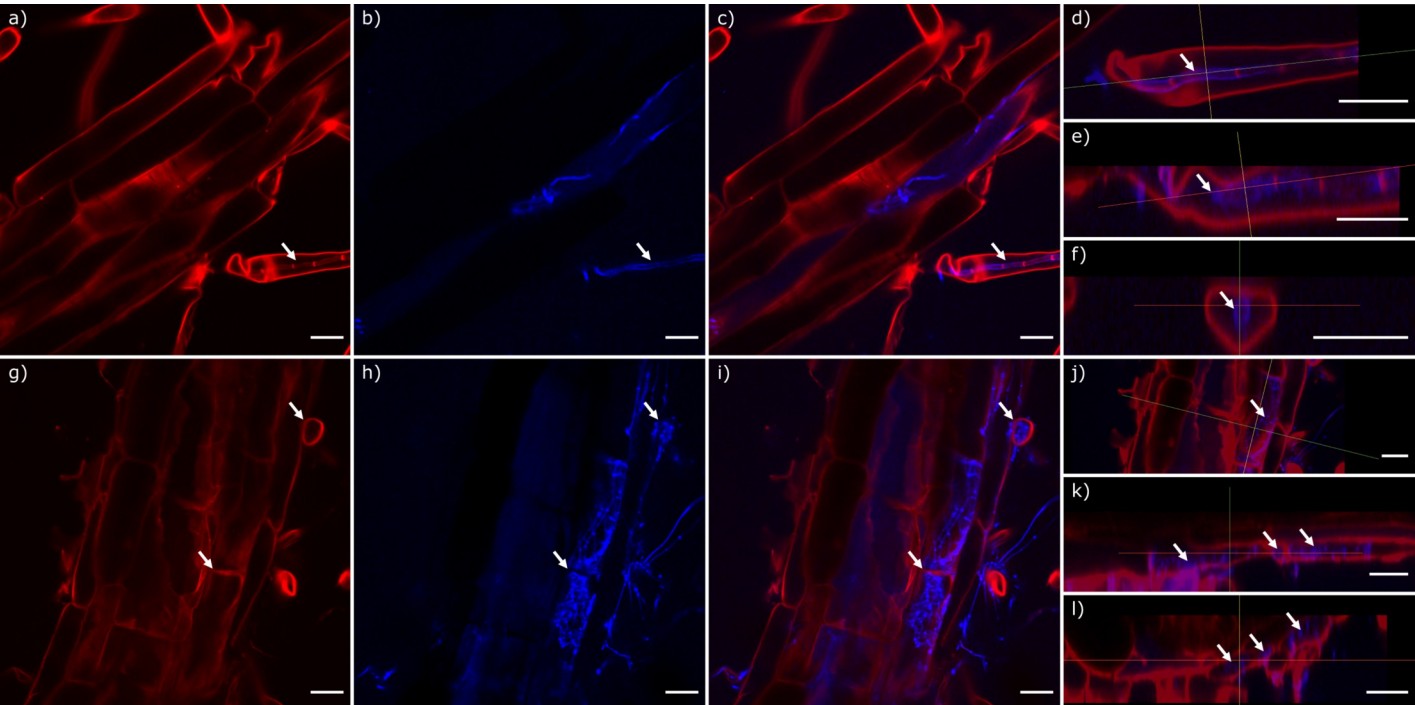

**Fig 5. Colonization of Arabidopsis roots by *Linnemannia elongata*. a-f**) NVP64cu at 25 days post inoculation, **g-l**) NVP80cu at 23 days post inoculation. **a,g**) Fluorescence signal from calcofluor white M2R staining; **b,h**) signal from wheat germ agglutinin 640R staining **c-f,i-l**) merged fluorescence. **d-f,j-l**) Orthogonal z-stack projections of root micrographs. **a-l**) White arrows indicate plant wall structures showing hyphae (blue) contained within intracellular spaces by plant cell walls (red). Scale bars represent 20 μm.

response to fungus" (GO:0050832). Down regulated genes were enriched for "response to extracellular stimulus" (GO:0009991) and "response to toxic substance" (GO:0009636). Broadly, these functional enrichments suggest external stimuli pathways were highly fluctuating in response to fungal treatment.

## Discussion

In this study, root symbiosis between *Arabidopsis thaliana* and *Linnemannia elongata* were characterized at the gene expression level and plants were phenotype for aerial plant growth and seed production. We were also able to compare the impact of strain and endosymbiont (BRE vs. MRE) on plant-fungal interactions since the two different *L. elongata* strains used harbored a different endosymbiont. Finally, we used RNA-seq to identify plant genes that are differentially expressed during Arabidopsis-*L. elongata* symbiosis in order to begin describing molecular mechanisms of interaction associated with plant growth promotion.

### *Linnemannia elongata* promotes Arabidopsis growth independent of endobacteria

This is the first study to explicitly test the impact of endobacteria on *Linnemannia*-plant associations. We found that *L. elongata* increased aerial plant growth compared to uninoculated controls, irrespective of the presence of endobacterial and independent of harvesting before or after flowering. These growth promotion effects of *Linnemannia* are corroborated by recent studies of *L. elongata* inoculated maize, where *L. elongata* increased the height and dry aerial

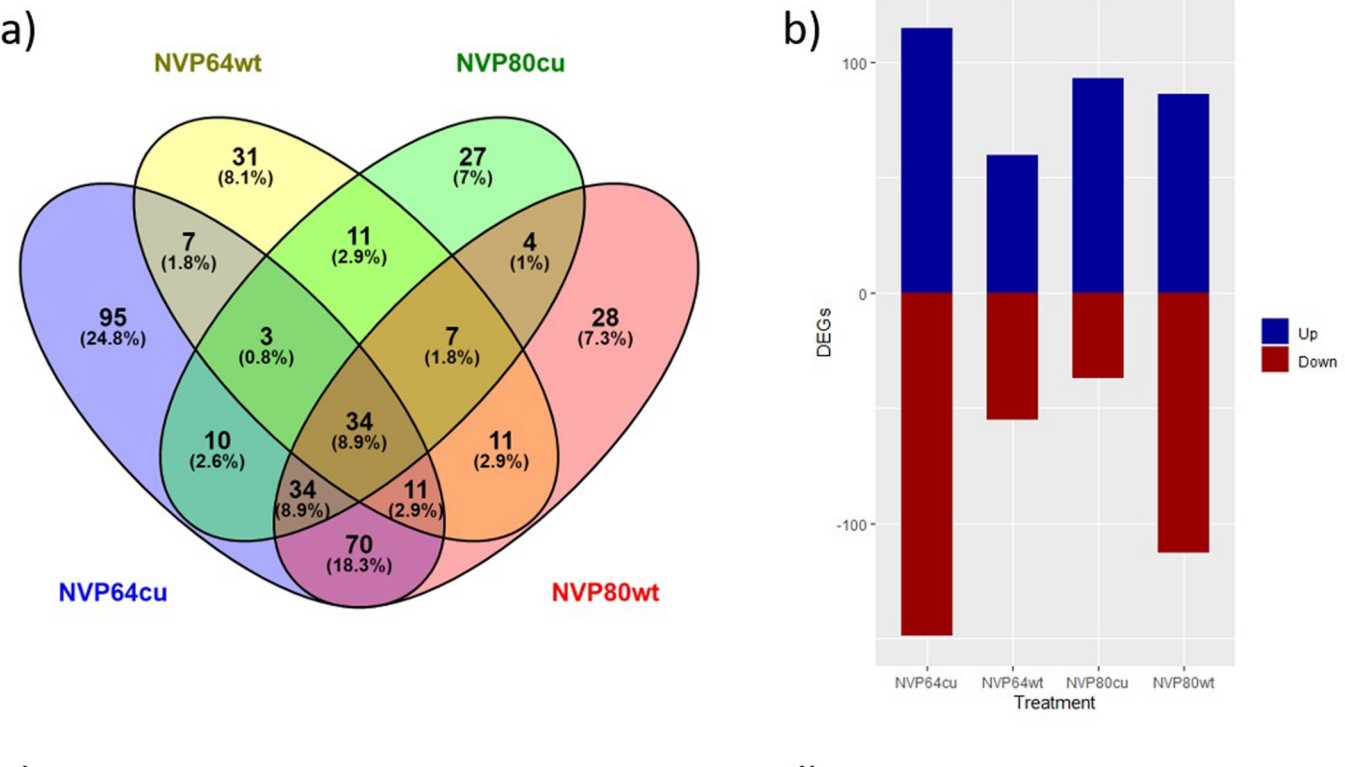

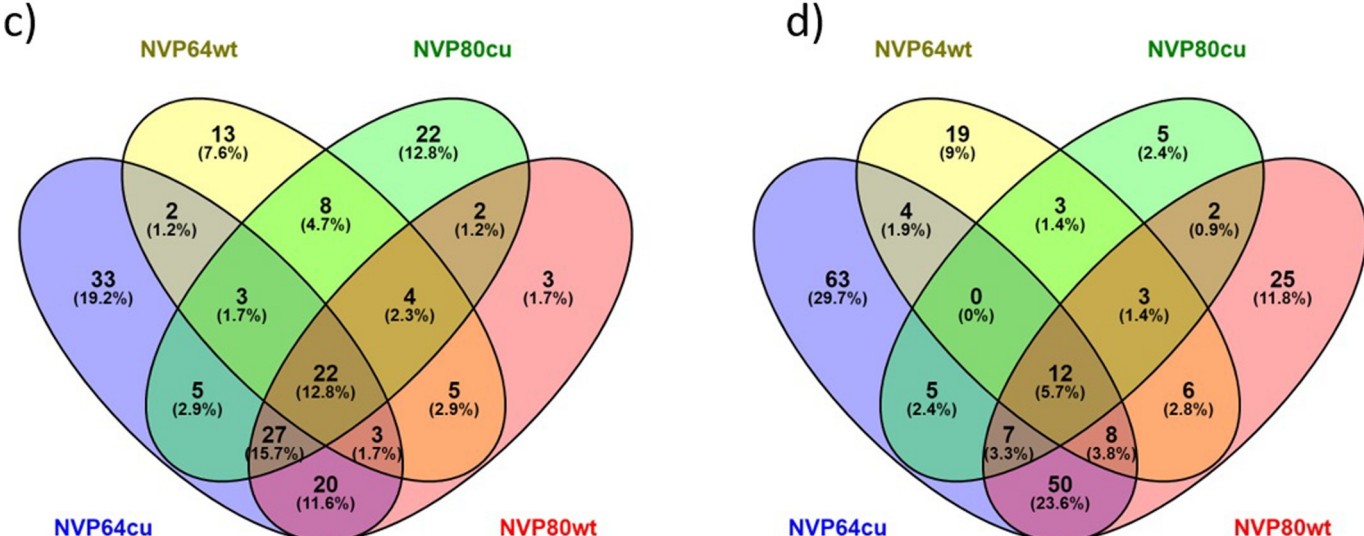

**Fig 6. Abundance of differentially expressed Arabidopsis genes.** *Arabidopsis thaliana* genes differentially expressed (DEGs) in roots colonized with *Linnemannia elongata* as compared to the uninoculated control, identified using DESeq2 with fold-change threshold of 1.5 and p-value threshold of 0.05. a) A Venn diagram of all DEGs in the final, filtered dataset. b) A bar graph of all DEGs, split between up- and down-regulated. c-d) Venn diagrams of c) up- and d) down-regulated DEGs identified for each fungal treatment.

biomass of maize in V3-V5 early vegetative stages, which corresponds to when maize has begun relying on photosynthesis and the environment for resources, rather than seed resources [1, 50]. Previous studies have reported that both MRE and BRE infection negatively impacts the growth of *Linnemannia*, thus, it is interesting that neither BRE nor MRE had a significant impact on plant growth in either experimental system [19, 20]. However, NVP80wt

**Table 2. Subset of Arabidopsis genes differentially expressed in response to Linnemannia elongata.**

| Functional Annotation | | | Log2 Fold-Change | | | | Name | Gene |
|---|---|---|---|---|---|---|---|---|
| **Broad** | **Middle** | **Detail** | **NVP 64cu** | **NVP 64wt** | **NVP 80cu** | **NVP 80wt** | | |
| Abiotic Stress | Hypoxia/ Oxidative Stress | Peroxidase superfamily protein | 2.26 | 1.92 | 1.84 | 1.98 | PER28 | AT3G03670 |
| | | Stachyose synthase, Raffinose synthase 4 | 1.72 | 0.93 | 1.31 | 1.54 | STS | AT4G01970 |
| Defense | Bacteria | Leucine-rich receptor-like protein kinase family protein | 1.5 | | 0.94 | 1.18 | FLS2 | AT5G46330 |
| | | Calcium-binding EF hand family protein | 1.38 | 1.57 | 1.78 | 1.63 | CML12 | AT2G41100 |
| | Fungus | Chitinase family protein | 3.02 | | 1.98 | 2.36 | F18O19.27 | AT2G43620 |
| | | homolog of RPW8 3 | 1.08 | | 1.1 | 1.04 | HR3 | AT3G50470 |
| | | SBP (S-ribonuclease binding protein) family protein | -4.15 | -3.1 | -3.2 | -3.49 | dl4875c | AT4G17680 |
| | | CAP (Cysteine-rich secretory proteins, Antigen 5, & Pathogenesis-related 1 protein) superfamily protein | 2.77 | 2.17 | 3.08 | 2.63 | CAPE3 | AT4G33720 |
| Development | Growth | promotes cell growth in response to light | 0.96 | 0.7 | 1.16 | 0.77 | LSH10 | AT2G42610 |
| | | xanthine dehydrogenase 2 | 0.82 | 1 | 0.91 | 0.75 | XDH2 | AT4G34900 |
| | Root | thalianol hydroxylase cytochrome P450, family 708, subfamily A, polypeptide 2 | -1.16 | | | -1.19 | THAH | AT5G48000 |
| | | Thalianol synthase 1 | -1.93 | | | -1.71 | THAS | AT5G48010 |
| | | marneral oxidase | -0.76 | -1.05 | -0.87 | -0.92 | MRO | AT5G42590 |
| | | Thalian-diol desaturase cytochrome P450, family 705, subfamily A, polypeptide 5 | -1.65 | -1.38 | -1.42 | -1.64 | THAD1 | AT5G47990 |
| Hormone Signaling | Auxin | Nitrilase 1 | -0.77 | | | -0.78 | NIT1 | AT3G44310 |
| | | nitrilase 2 | -1.05 | | -0.8 | -0.87 | NIT2 | AT3G44300 |
| | Brassinosteroid | squalene monooxygenase 2 | -1.8 | -1.76 | | -2.13 | SQE4 | AT5G24140 |
| | | baruol synthase 1 | 3.8 | | 2.74 | 3.44 | BARS1 | AT4G15370 |
| | Eth/JA | ethylene response factor | 0.74 | 0.76 | 0.69 | 0.8 | ERF59 | AT1G06160 |
| | | Integrase-type DNA-binding superfamily protein | 1.94 | | | 2.04 | TDR1 | AT3G23230 |
| | | ethylene-activated signaling pathway | 1.49 | | | 1.44 | RAP2.9 | AT4G06746 |
| | | ETHYLENE RESPONSE 2 | 0.84 | | 0.79 | 0.84 | ERT2 | AT3G23150 |
| | | 1-amino-cyclopropane-1-carboxylate (ACC) synthase 7 | -1.01 | -0.99 | -0.85 | -1.08 | ACS7 | AT4G26200 |
| | Signaling | cell wall-associated kinase | | 2.86 | 3.14 | | WAK1 | AT1G21250 |
| | | wall-associated kinase 2 | 1.41 | 1.11 | 1.53 | 1.16 | WAK2 | AT1G21270 |

A subset of twenty five DEGs having particularly interesting gene function and consistent significance across multiple fungal treatments. Log 2 fold change (LFC) values were calculated by DESeq2 and filtered at |LFC| = log2(1.5) = 0.58 and adjusted p-value = 0.05. Table is organized first by functional annotation, then by direction of regulation, and finally by the number of fungal treatments in which the gene was differentially expressed.

(with MRE) did show a weak trend towards smaller plants than NVP80cu in the potting mix experiment (**Fig 2**).

Previous studies have shown the *L. elongata* increases *Calibrachoa* flower production [49]. We demonstrate here that *L. elongata* fungal treatments had a strong positive effect on Arabidopsis seed size and total seed number. This may be an important plant trait to consider when assessing fitness costs of plant-associated microbes. Both NVP80cu and NVP80wt treatments had significantly higher total seed number compared to uninoculated millet controls, however, this was not the case for NVP64 inoculated treatments indicating strain variation. Interestingly, uninoculated millet control plants had a higher average seed size compared to some of the fungal treatments (**Fig 3B**). While it is difficult to draw strong conclusions with so few replicates, fitness trade-offs in seed size and seed mass of plants growing in stressful environmental conditions compared to non-stressful conditions is an interesting topic for future work [51, 52].

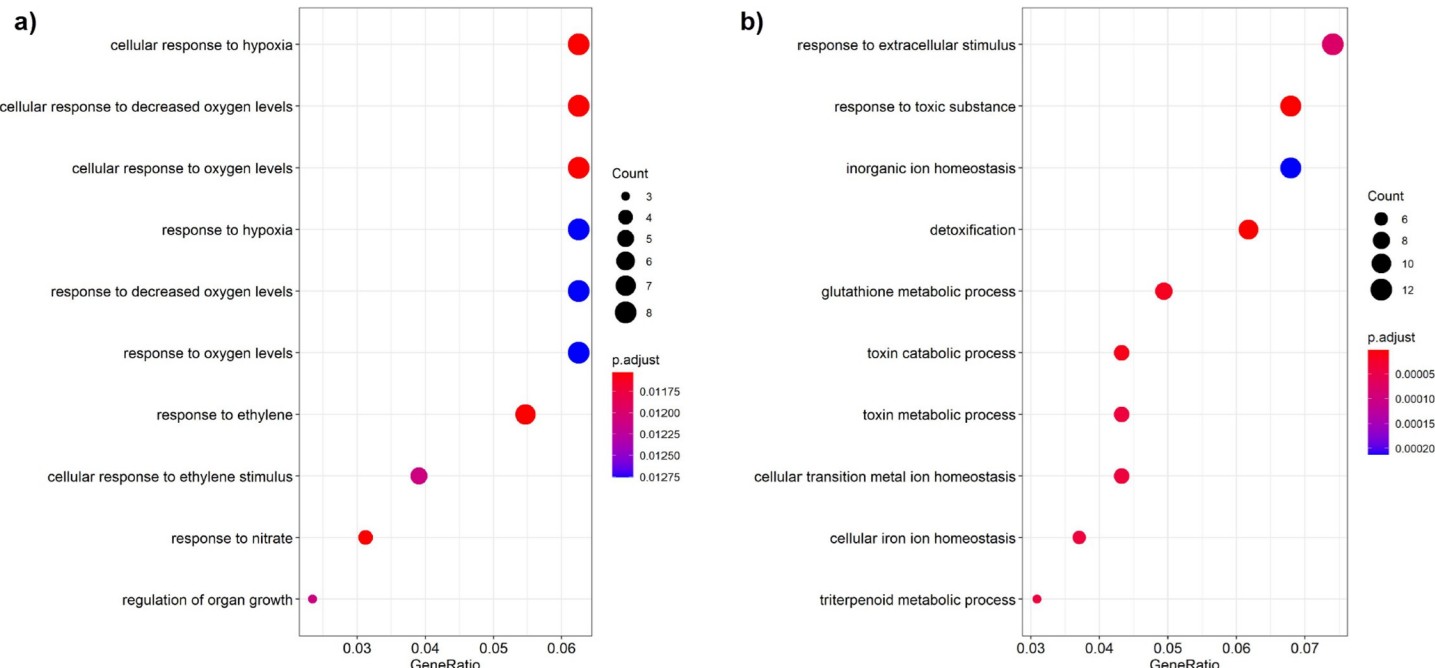

**Fig 7. GO enrichment of up and down regulated genes.** The ten GO categories with the strongest enrichment are displayed for both **a**) up and **b**) down regulated genes in response to fungal treatment. Color corresponds to the adjusted p-value according to the Benjamini-Hochberg Procedure while dot size corresponds to the number of differentially expressed genes matching a given GO category.

### *Linnemannia elongata* colonizes Arabidopsis root cells

Following extended co-culture of Arabidopsis and *L. elongata*, hyphae of *L. elongata* were observed to colonize root cells of Arabidopsis intracellularly. The intracellular hyphae were contained within epidermal cells, including those both with and without root hairs (**Fig 5**). Stained roots appeared healthy, but we are unable to determine if the root cells containing hyphae were still alive, or any distinctive function of intracellular hyphae. Intracellular hyphae are especially known from arbuscular mycorrhizal fungi, which produce highly branched arbuscules within the root cells of their host and provide an extensive exchange surface for nutrients [53]. As opposed to hyphae contained solely to the apoplast, these intracellular hyphae suggest a more intimate relationship between the host and fungus, which could allow for the exchange of nutrients, phytohormones, or other metabolites. Interesting, the plant growth promoting root endophyte *Serendipita* (*Piriformospora*) *indica* requires dead root cells for entrance into roots, but still provides benefits to the host [54]. However, although *Serendipita* and *Linnemannia* may both be found in plant roots, these fungi are phylogenetically distant, as *Serendipita* are Basidiomycetes while *Linnemannia* belong to Mucoromycota. Without further experiments to characterize nutrient exchange through these intracellular structures, it is not possible to ascribe structure with function [55].

### *Linnemannia elongata* may regulate Arabidopsis defense and abiotic stress responses

Up-regulated gene enrichments allow us to speculate on the transcriptional response of *A. thaliana* to *Linnemannia*. Several genes in the peroxidase superfamily III were upregulated in response to fungal inoculation. Nearly all these genes contain a signal peptide tagging them for export out of the cell [56]. Indeed, some are known to be involved in cell wall remodeling [57,

58], a process that must occur to establish mutualism. However, they are also involved in defense responses against pathogens. We propose a few alternative hypotheses why previously described defense responses are upregulated in a mutualistic interaction. First, the assigned gene ontology may inaccurately reflect the true function of these genes. They may be involved in mutualism, but if it was not previously shown, these genes will lack that GO term. Second, the upregulation of genes involved in defense response might be a priming response by *A. thaliana* as previously shown by Johnson et al. [15].

A number of plant hormones mediate the initiation and maintenance of plant-microbe symbioses, including auxins (most commonly IAA), jasmonates/jasmonic acid (JA), salicylic acid (SA), abscisic acid (ABA), ethylene (ET), and brassinosteroids (BRs). These hormones can be produced by both the plant and microbial symbionts and are often required to appropriately suppress and redirect the plant defense response in order for the microbe to establish symbiosis. The regulation and importance of each hormone is specific to the type of interaction (e.g. pathogen vs. mutualists) as well as the species of plant and microbe that are interacting. For example, ethylene suppresses AMF colonization, but promotes EM colonization [59–61]. Similarly, the beneficial non-mycorrhizal fungi *L. hyalina* and *Serendipita* (= *Piriformospora*) *indica* stimulate plant production of jasmonic acid and salicylic acid, respectively, when initiating symbiosis with Arabidopsis [62, 63]. While this study did not include direct measurement of phytohormone levels, we did identify several DEGs related to the biosynthesis and signaling of ethylene, auxin, and abscisic acid, which are discussed below.

**Root development and auxin.** We observed many upregulated genes in response to fungal inoculation that were previously shown to be upregulated during root development. This is interesting since development and stress response pathways overlap in plants [64, 65]. Many fungi synthesize and secrete auxin, a hormone well known to impact plant growth. *Podila verticillata* (= *Mortierella verticillata*) and *M. antarctica* both synthesize IAA and were shown to improve winter wheat seedling growth [18]. The genome of *L. elongata* (strain AG77) has the key genes of IAA synthesis and maize roots inoculated with *L. elongata* had a 37% increase in IAA concentration [1]. Our study found that *L. elongata* suppressed Arabidopsis auxin biosynthesis genes (*NIT2* and *GH3.7*), but up-regulated several auxin-responsive genes. Given that 1) Arabidopsis auxin biosynthesis is being down-regulated, 2) auxin synthesis is generally self-inhibitory in plants, and 3) auxin response genes are up-regulated, we hypothesize that the Arabidopsis roots are responding to *L. elongata*-derived auxins [66]. However, Arabidopsis auxin-related genes did not respond to initial or established *L. hyalina* colonization, even though Arabidopsis roots had a 3-fold increase in IAA concentration during initial colonization as compared to control roots [15, 62]. Moreover, some IAA was of fungal origin, as *L. hyalina* mycelium alone had a significantly higher IAA concentration than the tested pathogenic fungi [62]. It is worth noting that their assay did show a very brief response to auxin that quickly dissipated to background gene expression levels. Since we found increased auxin responsive gene expression during well-established symbiosis, our data indicate *L. elongata* may employ a different phytohormone regulatory strategy compared to *L. hyalina*.

Enhanced aerial plant growth by auxin-producing microbes is attributed to improved root structure, particularly lateral root growth, but assessing the impact of Mortierellaceae fungi on plant root development is not so straightforward [1]. Johnson et al. [15] found that *L. hyalina* had a slight, but significant negative impact on Arabidopsis root dry biomass compared to uninoculated plants; they identified three root development (*SHR*, *CPC*, and *AHP6*) genes that responded to *L. hyalina* as opposed to the plant pathogen *Alternaria brassicae*. These genes were not among the DEGs identified in our study. However, we did find that the entire operon-like gene set related to thalianol biosynthesis and metabolism (*MRN1*, *MRO*, *THAS1*, *THAH*, and *THAD*) was downregulated by *L. elongata* [67–70]. Thalianol-related metabolites

are predicted to function in promoting root development, but the mechanism is still under investigation [67]. Future research is needed to determine how each of these fungi impact Arabidopsis root development and how that relates to increased aerial plant growth.

**Ethylene (ET).** Ethylene is a plant hormone involved in maturation, senescence, and response to biotic and abiotic stress. Decreasing the level of ethylene in plant tissues generally promotes plant growth. The role of ET in plant response to pathogens is well characterized and includes increased ET biosynthesis and signaling through a single conserved pathway, which includes proteins in the *TDR1* family [71]. However, the origin and role of ET in the initiation of beneficial plant-fungal symbioses is specific to the fungi involved. For instance, elevated ET appears to promote colonization by ectomycorrhizal fungi, but inhibits colonization by AMF [59–61]. Moreover, the ET signaling pathway is known to have multiple points of crossover with other hormone signaling pathways, including JA and cytokinin, some of which occur through the ERF family of transcription factors, including TDR1 [71]. In our study, we found that Arabidopsis colonized by *L. elongata* down-regulated ACC synthase *ACS7*, which synthesizes the metabolite 1-amino-cyclopropane-1-carboxylate (ACC), a precursor of ethylene. However, some genes related to ethylene signaling were up-regulated in response to *L. elongata*. Since ET biosynthesis is downregulated in Arabidopsis roots in response to *L. elongata*, it is possible that related response genes are up-regulated via other hormone pathways. There were only three DEGs specifically associated with JA signaling in our dataset but they were each significant in only one fungal treatment.

**Abscisic acid (ABA), abiotic stress, & reactive oxygen species (ROS).** In general, we found that genes related to ABA and abiotic stress are down-regulated by *L. elongata*. These include the ABA synthesis enzyme *NCED3* and responses to drought, cold, salt, iron deficiency, potassium deficiency, phosphorus deficiency, and heavy metals [72]. Many plant growth promoting fungi are thought to transport water and nutrients to plants, particularly phosphorus. Mortierellaceae species are known to solubilize phosphate and improve phosphorus uptake in plants [73]. Considering the availability of potassium, phosphorus, and iron in the PNM growth medium, it is striking that so many genes related to deficiencies of these nutrients were down-regulated compared to the control plants. We suspect that the fungal mycelial network may have increased the availability of these nutrients to plant roots, by serving as an extension of the root system. There were two main exceptions to the reduction in abiotic stress gene expression: oxidative stress responses and a group of RmlC-like cupins superfamily proteins whose function is unknown.

ROS are a common plant defense response to both beneficial and pathogenic microbes [74]. Both *L. hyalina* and *L. elongata* stimulate ROS-responsive genes, although the two ROS-responsive genes specifically tested by Johnson et al. [15] were not among the DEGs in our dataset. Six of the up-regulated oxidative stress genes were peroxidases. One was a raffinose synthase. Raffinose is thought to act as an osmoprotectant and ROS scavenger [75]. Finally, we observed down-regulation of uridine diphosphate glycosyltransferase *UGT74E2*, which responds to ROS and drought to convert the auxin IBA to IBA-Glc [76]. ROS also stimulates conversion of IAA to IBA. Increased expression of *UGT74E2* further sequesters IBA and prevents oxidation back to IAA [76]. While no *UGT74E2* suppression or deletion mutant phenotypes have been reported, overexpression of *UGT74E2* leads to increased sensitivity to ABA [77]. In summary, we observe down-regulation of auxin synthesis, ABA synthesis and signaling, and an important gene connecting the ROS, ABA, and auxin pathways. From this, we infer that *L. elongata* stimulates ROS responsives genes, but these responses are isolated from other hormone pathways and limited to peroxidases and antioxidants.

**Calcium signaling and plant defense.** In addition to hormones, many plant-microbe interactions involve calcium signaling in plant roots [78]. *L. hyalina* symbiosis with

Arabidopsis is activated by calcium signaling [15]. Calcium-signaling was required for the plants to receive pathogen protection by *L. hyalina*, and for *L. hyalina* to colonize Arabidopsis roots; however, signaling-deficient plants still showed the wild-type aerial growth promotion. This suggests a calcium-signaling dependent defense response to limit the rate of root colonization by *L. hyalina*. Johnson et al. [15] identified four Ca-signaling genes (At3g47480, At3g03410, At5g23950, and At3g60950) that specifically responded to *L. hyalina*, as opposed to the plant pathogen *Alternaria brassicae*. These genes were not among the DEGs identified in our study. However, our RNA-seq experiment did demonstrate an up-regulation of the calcium-signaling gene *CML12*, which is induced by both stress and hormone signals, including auxin, touch, darkness, oxidative stress, and herbivory [79, 80].

DEG analyses indicate that *L. elongata* stimulated several general, fungal, and bacterial defense-related genes in Arabidopsis roots. However, we also noted suppression of genes involved in programmed cell death and production of defensin-like proteins meant to kill cells of invading organisms. As such, these defense responses could indicate both regulation of *L. elongata* colonization and a priming of the plant innate immune response, explaining the elevated expression of definitively bacterial defense genes like *FLS2*. As noted in maize-*L. elongata* symbioses, *L. elongata* may curate auxin levels to colonize maize roots and suppress systemic defense through the salicylic acid pathway [1]. Further, this active microbial regulation of the plant immune response may promote plant growth in a field environment by limiting further resource allocation to defense when attacked by pathogens [1, 81].

## Challenges of plant-fungi experiments

There are challenges to introducing fungi to plants without simultaneously altering other factors. Experiments carried out in potting mixes reiterate that uninoculated grains in control treatments not only invite colonization by environmental contaminants, but the grains themselves may introduce a strong and consistent negative impact on plant growth. However, we found the potting mix experiment was technically sufficient to collect data about seed production, while the agar inoculation approach allowed for more controlled growth. Now that *L. elongata* has been verified to impact plant growth under these conditions, more extensive experiments can be justified to further explore plant-fungal interactions. The more controlled agar system is well suited for high-throughput assays of plant and fungal knock-out mutants to further isolate important genes and pathways involved in this symbiosis, and for assaying early life-stage aerial growth and root gene expression. However, semi-solid rooting conditions may not be representative of plants grown in soil or more real-world conditions. An improved potting mix system based upon a grain-free inoculation protocol would be ideal to non-destructively track plant growth over time and to construct a more detailed description of how *L. elongata* affects plant growth and development.

## The role of phytohormones in fungi

While it is well established that fungi can manipulate and produce phytohormones, the effects of phytohormones on fungi are not well understood. Studies of plant hormone impacts on fungal growth and development are currently limited to a few plant pathogens and AM fungi. Exogenous ethylene is known to promote fungal spore germination and mycelial growth [82–84]. For example, exogenous ethylene is required for spore germination in species of *Alternaria*, *Botrytis*, *Penicillium*, and *Rhizopus* and often promotes mycelial growth [84]. It is worth noting that these fungal species infect fruit, and likely evolved through selection for spore germination in the presence of ripening fruit, limiting the relevance of those findings to mechanisms employed by root-associated fungi [83]. Gryndler et al. [85] found that exogenous auxin

(IAA) repressed hyphal growth of two AM fungi, *Glomus fistulosum* and *G. mosseae*, at biologically relevant concentrations, but abscisic acid (ABA) and cytokinins had no perceivable effect until applied in concentrations very high, non-physiologically relevant, concentrations. The current model of phytohormone regulation of AM fungi suggests that 1) SA inhibits pre-symbiotic growth; 2) ethylene, JA, and cytokinins inhibit symbiotic fungal growth inside plant roots; and 3) auxin, JA, and ABA promote the formation and function of arbuscules within plant root cells [86]. It is still unclear how these relationships and regulatory systems apply to the growth, development, or plant associations of *L. elongata*, but these are important questions to consider in future plant-fungal interaction studies.

## Conclusions

In conclusion, we phenotyped Arabidopsis at early and late life stages during a stable symbiosis with *Linnemannia* in soil and agar-based media. We demonstrated that *L. elongata* promotes Arabidopsis above-ground vegetative growth and seed production. This plant phenotype was found to be independent of whether *L. elongata* isolates were colonized by MRE or BRE endohyphal bacterial symbionts. Plant-fungal symbiosis functions appears to involve auxin production by the mycobiont and the stimulation of ethylene and ROS response pathways in the plant host. Future research is needed to test these hypotheses and further characterize the fungal side of this symbiosis.

## Supporting information

**S1 File. Seed counting, automated image analysis in ImageJ and protocol for chloroform DNA extraction with CTAB 2X buffer.**
(DOCX)

**S1 Fig. Arabidopsis seedlings used in plant-fungal interaction assays.** Panel **a**) 10 day old *Arabidopsis thaliana* seedlings on 1xMS germination plates and **b**) 11 day old Arabidopsis seedlings and blocks of media (colonized by fungi in fungal treatments or sterile in uninoculated control treatments) as arranged on PNM plates for the agar-based plant-fungal interaction experiments.
(TIF)

**S2 Fig. Seeds to be counted by image analysis.** *Arabidopsis thaliana* was grown to maturity and the seeds of each plant collected by Aracon tubes and stored in Eppendorf tubes. After careful cleaning of the seed to remove stems, petals, and other plant debris, approximately 14 mg of seeds per sample were weighed on an ultrasensitive balance, adhered to a piece of white paper using a glue stick, covered by clear packing tape, scanned, and counted by image analysis in ImageJ. **a**) The scanned image of the subsampled seeds. **b**) The image analysis output, with areas identified as a "seed" outlined in red.
(TIF)

**S3 Fig. Agar plates with Arabidopsis plants in the growth chamber.** *Arabidopsis thaliana* seeds germination and Arabidopsis-*L. elongata* interaction studies were conducted on agar plates. These were incubated in a Percival growth chamber. Plates were stacked on a gentle angle to encourage smooth directional root growth along the agar surface.
(JPG)

**S4 Fig. Bolting phenotype.** The arrow indicates the elongation of the *Arabidopsis thaliana* inflorescence away from the rosette of leaves which was considered to indicate "bolting."
(TIF)

**S5 Fig. Violin plots of Arabidopsis seed image area.** *Arabidopsis thaliana* was grown to maturity and the seeds collected by Aracon tubes. Facet names indicate the composition of the potting mix in which Arabidopsis plants were grown. The untreated control (NoMillet) contrasted treatments where the sterile potting soil was mixed 97:3 v:v with sterile millet mix (Uninoculated), or millet mix inoculated with one of four *Linnemannia elongata* strains (NVP64cu, NVP64wt, NVP80cu, or NVP80wt). A subset of seeds from 5 samples per treatment (sample indicated by 'Rep') were adhered to white paper and imaged using an Epson scanner. The y-axis indicates the pixel count of each individual seed scanned for each rep and treatment using ImageJ.
(TIFF)

**S6 Fig. Linnemannia elongata strains equivalently colonized Arabidopsis roots.** RNA was extracted from *Arabidopsis thaliana* roots colonized by *Linnemannia elongata*, pooled from all three plants on each agar plate, from three plates per treatment. The inferred ratio of fungal: plant cDNA is based on the qPCR results and standard curves for each qPCR primer set. Since Arabidopsis GADPH and *L. elongata* RPB1 are single copy genes, the ratio of fungal and plant template provides a normalized estimate of fungal colonization of plant roots.
(JPEG)

**S7 Fig. Principal component analysis of differential Arabidopsis gene expression.** *Arabidopsis thaliana* root RNAseq data analyzed using DESeq2, sequenced from three biological replicates taken from each of the uninoculated control and fungal treatments inoculated with *Linnemannia elongata*.
(TIFF)

**S8 Fig. Volcano plots of differential gene expression.** Pairwise comparisons of normalized *Arabidopsis thaliana* gene expression between fungal treatments and the uninoculated control, calculated from the DESeq2 analyses. Each point represents a gene, plotted by adjusted p-value and Log2 Fold Change (LFC) in expression between the fungal treatment and the control. Vertical dashed lines indicate the |LFC| = 1 threshold and horizontal lines indicate the adjusted p-value threshold of 0.05 used to identify genes with significant changes in expression. Genes are colored by which of the LFC and p-value cutoffs were exceeded: gray = failed both; green = exceeded only the LFC cutoff, but not the p-value cutoff; blue = exceeded p-value cutoff, but not LFC; red = exceeded both cutoffs.
(TIF)

**S1 Table. A map of the light levels in the growth chamber.** *Arabidopsis thaliana* seeds germination and Arabidopsis-*L. elongata* interaction studies were conducted on agar plates. These were incubated in a Percival growth chamber. Each shelf in the chamber was divided into nine regions and the light level in each region was measured using an LI-250A light meter (LI-COR) with the chamber door closed to ensure realistic conditions. Light levels on the middle and bottom shelves were measured after arranging agar plates on the above shelf/shelves.
(XLSX)

**S2 Table. qPCR primer sets.** The qPCR primer sets used to quantify fungal colonization of plant roots and check for BRE/MRE in cured and wild-type fungal strains and fungus-colonized plant roots.
(XLSX)

**S3 Table. Linear modeling of Arabidopsis aerial dry weight as a function of light level.** The aerial dry biomass of *Arabidopsis thaliana* plants harvested from agar-based Arabidopsis-*Linnemannia elongata* interaction experiments and modeled as a function of light level. Medium

indicates the composition of the medium on which *L. elongata* was cultured: KM = Kaefer Medium; MEA = Malt Extract Agar. Treatment indicates the strain of *L. elongata* with which Arabidopsis was inoculated or the uninoculated control.
(XLSX)

**S4 Table. Linear modeling of Arabidopsis aerial dry weight as a function of treatment and medium.** The aerial dry biomass of *Arabidopsis thaliana* plants from agar-based Arabidopsis-*Linnemannia elongata* interaction experiments, modeled as a function of treatment (control v. different strains of *Linnemannia elongata*), the medium on which the fungi had been cultured, and any interaction between those terms. We also conducted pairwise comparisons within treatments of the estimated marginal means (EMMs) for each inoculating medium.
(XLSX)

**S5 Table. Linear modeling of Arabidopsis aerial dry weight as a function of starting seedling root length.** The aerial dry biomass of *Arabidopsis thaliana* plants from agar-based Arabidopsis-*Linnemannia elongata* interaction experiments, modeled as a function of seedling starting root length. There were no significant differences in the slope of the relationship of starting root length to final aerial dry weight across experimental rounds or treatments.
(XLSX)

**S6 Table. qPCR of plant, fungal, and endobacterial genes from RNA.** Values indicate the mean (n = 2) qPCR cycle number at which amplification reached the threshold of detection ($C_t$) for each locus tested in each cDNA library from the *Arabidopsis thaliana* root RNA samples used in the RNAseq experiment. Dash = not tested. Arabidopsis GADPH and *Linnemannia elongata* RPB1 are single copy genes. The bacterial 16S gene is multicopy, which was necessary for detection, since these endobacteria are in very low abundance in the fungal hyphae.
(XLSX)

**S7 Table. Molecular results of RNA sequencing run.** The number of sequenced reads passing initial filtration by the sequencer, the percentage of those reads that mapped to the combined reference transcriptome, and the proportion of mapped reads that mapped to plant or fungal transcripts.
(XLSX)

**S8 Table. Arabidopsis genes differentially expressed in response to Linnemannia elongata.** Log 2 fold change (LFC) values were calculated by DESeq2 and filtered at |LFC| = log2(1.5) = 0.58 and adjusted p-value = 0.05. Table is organized first by functional annotation, then by direction of regulation, and finally by the number of fungal treatments in which the gene was differentially expressed.
(XLSX)

## Acknowledgments

We would like to thank Abigail Bryson and Bryan Rennick for their extensive assistance with setting up experiments, Xinxin Wang for assistance collecting Arabidopsis seeds from plant material, and Natalie Golematis for help with antibiotic passaging to cure fungal strains and DNA extractions for qPCR analyses. We would like to thank Dr. Zsofia Szendrei for generously providing access to her lab microbalance for weighing seeds and plants. We are grateful to Keith Koonter and Dr. Matthew Greishop for sharing their automated image analysis pipeline.

## Author Contributions

**Conceptualization:** Natalie Vandepol, Gregory Bonito.

**Data curation:** Natalie Vandepol, Alan Yocca, Jason Matlock.

**Formal analysis:** Natalie Vandepol, Alan Yocca, Jason Matlock.

**Funding acquisition:** Patrick Edger, Gregory Bonito.

**Investigation:** Julian Liber, Gregory Bonito.

**Methodology:** Natalie Vandepol.

**Project administration:** Patrick Edger, Gregory Bonito.

**Supervision:** Patrick Edger, Gregory Bonito.

**Validation:** Natalie Vandepol.

**Visualization:** Julian Liber.

**Writing – original draft:** Natalie Vandepol, Julian Liber, Alan Yocca, Gregory Bonito.

**Writing – review & editing:** Natalie Vandepol, Julian Liber, Alan Yocca, Jason Matlock, Patrick Edger, Gregory Bonito.

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
