## [Decision Letter · Decision Letter 0]

11 Feb 2022

PONE-D-21-37581Linnemannia elongata (Mortierellaceae) stimulates Arabidopsis thaliana aerial growth and responses to auxin, ethylene, and reactive oxygen speciesPLOS ONE

Dear Dr. Bonito,

Thank you for submitting your manuscript to PLOS ONE. After careful consideration, we feel that it has merit but does not fully meet PLOS ONE’s publication criteria as it currently stands. Therefore, we invite you to submit a revised version of the manuscript that addresses the points raised during the review process.

The paper has been submitted to te revision of two experts who both generally positively evaluated it. However both suggested some minor revisions that should be included in the paper in order to improve it and render acceptable for publication. Authors are invited to carefully follow suggestions from reviewers and modify the text according it.

We look forward to receiving your revised manuscript.

Kind regards,

Sabrina Sarrocco

Academic Editor

PLOS ONE

Journal Requirements:

“Funding sources: GB acknowledges support from the US National Science Foundation DEB 1737898 and the US Department of Agriculture NIFA MICL02416.”

We note that you have provided funding information within the Acknowledgements Section. Please note that funding information should not appear in the Acknowledgments section or other areas of your manuscript. We will only publish funding information present in the Funding Statement section of the online submission form.

 “GB and PE acknowledges support from the US National Science Foundation DEB 1737898. GB acknowledges support fromthe US Department of Agriculture NIFA MICL02416. The funders had no role in study design, data collection and analysis, decision to publish, or preparation of the manuscript.”

Reviewers' comments:

Reviewer's Responses to Questions

**Comments to the Author**

1. Is the manuscript technically sound, and do the data support the conclusions?

Reviewer #1: Yes

Reviewer #2: Yes

2. Has the statistical analysis been performed appropriately and rigorously? 

Reviewer #1: Yes

Reviewer #2: Yes

3. Have the authors made all data underlying the findings in their manuscript fully available?

Reviewer #1: Yes

Reviewer #2: Yes

4. Is the manuscript presented in an intelligible fashion and written in standard English?

Reviewer #1: Yes

Reviewer #2: Yes

5. Review Comments to the Author

Reviewer #1: This is an interesting study to examine the role of Linnemannia elongata (Mortierellaceae) and its endosymbionts in Arabidopsis thaliana aerial growth, although endosymbionts had no significant effects on Arabidopsis performance and the expression of genes in Arabidopsis in this study. Overall, the study is solid, and the manuscript is well written. However, there are a few areas of the manuscript that could be strengthened a little.

Abstract: it may be necessary to state the significance of the study at the end of abstract.

Introduction:

L47: Please add the refs.

M&M:

Please provide some information about two isolates of L. elongata, NVP64 and NVP80. For example, where did you isolate?

L143. Please correct the format of the ref.

L193. Please show the ingredients for the ratio.

L234. How many replicates/plates for each treatment in the agar-based experiment.

L324. What do you mean of “2x”?

L352. Superscript

L373. Please correct the format of ref.

L.376. You randomly selected five plants for extracting root RNA. Why are there only three biological replicates?

Results:

L408. Somehow, fungal treatment can offset the negative effect of millet on plant growth?

L414. Fig. 3a

L416-417. From Fig. 3b, it seems that uninoculated control also had slightly higher average seed mass than NVP64wt and NVP80cu.

L428. Plants?

L433. Please correct the format of ref.

L443-451. It is not clear where are these data from. Please cite figures or tables here.

L485. How did you get this? Try to use statistical analysis (e.g., PERMANOVA) to reveal it.

L508-509. Please show the specific GOs here and site Table 2.

Discussion:

L529. It is surprising that bacterial endosymbionts negatively impact the growth of Linnemannia. According to the previous study (Büttner et al., 2021), bacterial endosymbionts (Ca. Mycoavidus necroximicus) can protect Mortierella/ Linnemannia from the attack of micro-predators.

L540-544. This sentence is too long. Try to break it down into two sentences.

L555. This sentence is less connected to previous ones. Is Serendipita (Piriformospora) indica phylogenetically close to L. elongate?

L637. Refs?

L649. Although K, P, and Fe are available in PNM medium, there is a great demand for plant growth in the fungal treatment. This is probably the main reason for so many down-regulated genes related to nutrients.

L651. Correct the format of ref.

Reviewer #2: The manuscript by Vandepol and colleagues presents the results of several experiments aiming to demonstrate Linnemania elongate as effective plant growth promoting organism and verifying the role of endobacteria in this process. Apart of mRNA sequencing Authors performed a series of classical agar and pot experiments aiming to measure plant biomass and seed production. I appreciate presentation of results, showing no significant influence of EHB on the PGP. These are rarely demonstrated and highly valuable for the community. However, it is not clearly explained why Authors assume EHB may have such an effect on PGP processes. The study is a valuable contribution proving existence and describing mechanisms of PGP in Mortierellomycotina. Below I present some minor comments.

Lines 49-51: This sentence seems to be a far-reaching simplification. Especially, speaking about Mucoromycota it would be valuable to discuss at least mycorrhiza-like or paramycorrhiza concepts.

Line 60: I would remove “some of which are EM fungi” – see previous comment

Line 88: Arabidopsis once is italicized once without – please unify throughout the text

Line 486: “DESeq2 provided p-values for each comparison” – DESeq2 is software package you used to perform specific type of analysis, please reformulate here and further eg. line 492.

Line 540: I miss information on number of these replicates in methodological part. Please complete.

The used strains should be deposited in public culture collection.

6. PLOS authors have the option to publish the peer review history of their article (what does this mean?). If published, this will include your full peer review and any attached files.

Reviewer #1: No

Reviewer #2: No

---

## [Author Response · Author response to Decision Letter 0]

26 Feb 2022

We appreciate the reviewers for their time and comments. Below we provide a point-by-point response to the peer-reviews.

Reviewer 1

This is an interesting study to examine the role of Linnemannia elongata (Mortierellaceae) and its endosymbionts in Arabidopsis thaliana aerial growth, although endosymbionts had no significant effects on Arabidopsis performance and the expression of genes in Arabidopsis in this study. Overall, the study is solid, and the manuscript is well written. However, there are a few areas of the manuscript that could be strengthened a little. 

1. Abstract: it may be necessary to state the significance of the study at the end of abstract.

We’ve added a concluding significance statement: 

“Together, these results indicate that beneficial plant growth promotion and seed mass impacts of L. elongata on Arabidopsis are likely driven by plant hormone and defense transcription responses after plant-fungal contact, and that plant phenotypic and transcriptional responses are independent of whether the fungal symbiont is colonized by Mollicutes or Burkholderia-related endohyphal bacteria.”

2. Introduction : L47: Please add the refs.

New references have been included.

3. M&M: Please provide some information about two isolates of L. elongata, NVP64 and NVP80. For example, where did you isolate?

This has been included

4. L143. Please correct the format of the ref.

This has been done.

5. L193. Please show the ingredients for the ratio.

This has been added: “transplanted into 4 in3 pots with SureMix mixed with the appropriate millet mix treatment at 3% by volume.”

6. L234. How many replicates/plates for each treatment in the agar-based experiment.

This has been added “The number of biological replicates per treatment varied by experiment as follows: 27 for the bolting assay, 12 for the media panel, and 27 for the endobacteria panel.”

7. L324. What do you mean of “2x”?

We replaced with ‘twice’

8. L352. Superscript 

Fixed

9. L373. Please correct the format of ref.

Fixed

10. L.376. You randomly selected five plants for extracting root RNA. Why are there only three biological replicates? 

This was added to at LN311 to clarify: ‘three of which were selected as triplicate biological replicates for RNA sequencing based on having aerial dry weights closest to the mean for that treatment’

11. Results:L408. Somehow, fungal treatment can offset the negative effect of millet on plant growth?

This is correct, and is elaborated on further in the paragraph.

12. L414. Fig. 3a

fixed

13. L416-417. From Fig. 3b, it seems that uninoculated control also had slightly higher average seed mass than NVP64wt and NVP80cu.

This is reported on Ln 423: “Unlike total seed mass, the average seed mass of the Uninoculated control was slightly higher than NVP80wt and NVP64cu, but not significantly different from the NoMillet control (Fig. 3b)”. This pertains to the average mass of each seed (total seed mass/number of seeds).

14. L428. Plants?

We removed ‘fungal’ so it now reads ‘treatments’, which refers to the inoculum used (NVP64+, NVP64-, NVP80+, NVP80-, control).

15. L433. Please correct the format of ref.

fixed

16. L443-451. It is not clear where are these data from. Please cite figures or tables here.

Analysis of variance analyses were used to determine which factors were significant for inclusion in the linear models. Following standard convention, non-significant factors based on ANOVAs are reported only in the text directly.

17. L485. How did you get this? Try to use statistical analysis (e.g., PERMANOVA) to reveal it.

Thank you for suggesting we test this. Initially, this was an observation that there was no clustering based on the first two principal components in 2-dimensional space. Applying a PERMANOVA test in R (code shown at the bottom of the rnaseq.R file) revealed there was no significant clustering based on either isogenic strain or cured/wild-type.

18. L508-509. Please show the specific GOs here and site Table 2.

We have included specific GO numbers here and have cited table 2.

19. Discussion: L529. It is surprising that bacterial endosymbionts negatively impact the growth of Linnemannia. According to the previous study (Büttner et al., 2021), bacterial endosymbionts (Ca. Mycoavidus necroximicus) can protect Mortierella/ Linnemannia from the attack of micro-predators. 

The negative impact of endosymbionts on Linnemannia growth in pure culture was previously published by Uehling et al 2017 and Desiro et al. 2018. It was not assessed in the Büttner et al., 2021 paper. 

20. L540-544. This sentence is too long. Try to break it down into two sentences.

This sentence has been rewritten. “While it is difficult to draw strong conclusions with so few replicates, fitness trade-offs in seed size and seed mass of plants growing in stressful environmental conditions compared to non-stressful conditions is an interesting topic for future work [47,48].”

21. L555. This sentence is less connected to previous ones. Is Serendipita (Piriformospora) indica phylogenetically close to L. elongate?

Thank you. The sentence has been better connected by adding this transition.

“However, although Serendipita and Linnemannia may both be found in plant roots, these fungi are phylogenetically distant, as Serendipita are Basidiomycetes while Linnemannia belong to Mucoromycota. Without further experiments to characterize nutrient exchange through these intracellular structures, it is not possible to ascribe structure with function [55].”

22. L637. Refs?

Refs added

23. L649. Although K, P, and Fe are available in PNM medium, there is a great demand for plant growth in the fungal treatment. This is probably the main reason for so many down-regulated genes related to nutrients. 

We have revised this sentence Ln643: “Considering the availability of potassium, phosphorus, and iron in the PNM growth medium, it is striking that so many genes related to deficiencies of these nutrients were down-regulated compared to the control plants. The fungal mycelial network may have increased the availability of these nutrients to plant roots, by serving as an extension of the root system.”

24. L651. Correct the format of ref.

This has been fixed

---

## [Editor Report · Decision Letter 1]

1 Mar 2022

Linnemannia elongata (Mortierellaceae) stimulates Arabidopsis thaliana aerial growth and responses to auxin, ethylene, and reactive oxygen species

PONE-D-21-37581R1

Dear Dr. Bonito,

We’re pleased to inform you that your manuscript has been judged scientifically suitable for publication and will be formally accepted for publication once it meets all outstanding technical requirements.

Kind regards,

Sabrina Sarrocco

Academic Editor

PLOS ONE
---

## [Editor Report · Acceptance letter]

10 Mar 2022

PONE-D-21-37581R1 

*Linnemannia elongata* (Mortierellaceae) stimulates *Arabidopsis thaliana* aerial growth and responses to auxin, ethylene, and reactive oxygen species 

Dear Dr. Bonito:

I'm pleased to inform you that your manuscript has been deemed suitable for publication in PLOS ONE. Congratulations! Your manuscript is now with our production department. 

Kind regards, 

on behalf of

Dr Sabrina Sarrocco 

Academic Editor

PLOS ONE